# Towards Practical Alternating Least-Squares for CCA

**Zhiqiang Xu and Ping Li**

Cognitive Computing Lab
Baidu Research
No.10 Xibeiwang East Road, Beijing, 10085, China
10900 NE 8th St, Bellevue, WA 98004, USA
{xuzhiqiang04,liping11}@baidu.com

## Abstract

Alternating least-squares (ALS) is a simple yet effective solver for canonical correlation analysis (CCA). In terms of ease of use, ALS is arguably practitioners' first choice. Despite recent provably guaranteed variants, the empirical performance often remains unsatisfactory. To promote the practical use of ALS for CCA, we propose **truly alternating least-squares**. Instead of approximately solving two independent linear systems, in each iteration, it simply solves two coupled linear systems of half the size. It turns out that this coupling procedure is able to bring significant performance improvements in practical setting. Inspired by the accelerated power method, we further propose **faster alternating least-squares**, where momentum terms are introduced into the update equations. Theoretically, both algorithms enjoy linear convergence rate. To make faster ALS even more practical, we put forward **adaptive alternating least-squares** to avoid tuning the momentum parameter, which is as easy to use as the plain ALS while retaining advantages of the fast version. Experiments on several datasets empirically demonstrate the superiority of the proposed algorithms to several recent variants of CCA solvers.

## 1 Introduction

Canonical correlation analysis [11] is a classical statistical tool for finding directions of the maximal correlations between data sources of the same phenomenon, which has found widespread applications in high-dimensional data analysis such as regression [12], clustering [5], classification [13], and word embedding [7], to name a few. Let $\mathbf{X} \in \mathbb{R}^{d_x \times n}$ and $\mathbf{Y} \in \mathbb{R}^{d_y \times n}$ be the data matrices[1] of two views with empirical cross-covariance matrix and two auto-covariance matrices given by

$$\mathbf{C}_{xy} = \frac{1}{n}\mathbf{X}\mathbf{Y}^\top, \ \ \mathbf{C}_{xx} = \frac{1}{n}\mathbf{X}\mathbf{X}^\top + r_x\mathbf{I}, \ \ \mathbf{C}_{yy} = \frac{1}{n}\mathbf{Y}\mathbf{Y}^\top + r_y\mathbf{I},$$

respectively, where $r_x$ and $r_y$ are positive regularization parameters for avoiding ill-conditioned matrices and $\mathbf{I}$ represents the identity matrix of the appropriate size. CCA aims to find projection matrices $\mathbf{\Phi} \in \mathbb{R}^{d_x \times k}$ and $\mathbf{\Psi} \in \mathbb{R}^{d_y \times k}$ such that the cumulative correlation between two views is maximized after the projection of each view [19, 16]:

$$\max_{\mathbf{\Phi}^\top \mathbf{C}_{xx} \mathbf{\Phi} = \mathbf{\Psi}^\top \mathbf{C}_{yy} \mathbf{\Psi} = \mathbf{I}} \mathrm{tr}(\mathbf{\Phi}^\top \mathbf{C}_{xy} \mathbf{\Psi}). \tag{1}$$

It is well-known that the global optimum to Problem (1), which is also known as the canonical subspaces, can be obtained via a $k$-truncated singular value decomposition (SVD) on the whitened

cross-covariance matrix $\mathbf{C} = \mathbf{C}_{xx}^{-\frac{1}{2}}\mathbf{C}_{xy}\mathbf{C}_{yy}^{-\frac{1}{2}}$, i.e.,

$$(\mathbf{U}, \mathbf{V}) = (\mathbf{C}_{xx}^{-\frac{1}{2}}\mathbf{P}, \mathbf{C}_{yy}^{-\frac{1}{2}}\mathbf{Q}), \tag{2}$$

where $\mathbf{P}$ and $\mathbf{Q}$ are the top-$k$ left and right singular subspaces of $\mathbf{C}$. Simply applying the partial SVD of $\mathbf{C}$ by inverting matrices $\mathbf{C}_{xx}$ and $\mathbf{C}_{yy}$, however, is computationally prohibitive for high-dimensional datasets, as the complexity of matrix inversions can be as high as $O(d^3)$, where $d = \max\{d_x, d_y\}$, and the data sparsity of $\mathbf{X}$ and $\mathbf{Y}$ can not be utilized then.

To address this computational issue of CCA, there have been a range of relevant algorithms proposed recently in different settings [22, 15, 16, 10, 20, 1, 8, 2, 6, 4]. In this work, we focus on the block and off-line setting where $k > 1$ and the collection of instance pairs, i.e., $(\mathbf{X}, \mathbf{Y})$, is ready. In terms of ease of use, in this setting, alternating least-squares (ALS) [20, 10] is arguably the first choice from a user's perspective, by virtue of the simplicity, the fewest parameters, and guaranteed convergence. Nonetheless, as we will see in our experiments, its effectiveness, especially solutions of high accuracy, often comes at the cost of slow convergence. Particularly, [20] considered inexact alternating least-squares for the vector case $k = 1$. However, in order for the block case to work, one has to set the block size to $2k$ rather than $k$ and needs a post-processing step to randomly project the resulting solution of $2k$-dimensional subspace onto a $k$-dimensional subspace, as is demonstrated in [10]. The update equations of alternating least-squares in both [20] and [10] are derived from the power method on an augmented real symmetric matrix, i.e., $\mathbf{A} = \begin{pmatrix} & \mathbf{C}_{xy} \\ \mathbf{C}_{xy}^\top & \end{pmatrix}$. However, the power method can only find top eigenspaces corresponding to the largest eigenvalues in magnitude rather than the real part. Given the special eigen-structure of $\mathbf{A}$ [23, 20, 10], the block size has to be at least $2k$ to recover a top-$k$ canonical subspace $(\mathbf{U}, \mathbf{V})$. It is clear that this way that the block CCA solver proceeds not only causes a significant increase in both time and space, but may also degrade the quality of the final solution due to the random projection.

Thus, the question one would naturally ask is:

*Is there any variant of ALS that is able to recover $(\mathbf{U}, \mathbf{V})$ with block size $k$?*

In this paper, we offer a simple answer in the affirmative. Recall that the power iteration in [20, 10] leads to simultaneous approximations to exact iterates $\mathbf{\Phi}_t^\star$ and $\mathbf{\Psi}_t^\star$ on two canonical variables and ends up solving two independent linear systems. What we are going to change here is to do sequential approximations to $\mathbf{\Phi}_t^\star$ and $\mathbf{\Psi}_t^\star$ with block size $k$, arriving at an algorithm that approximately solves two *coupled* linear systems of *half the size* per iteration (see Algorithm 1). To stress the difference, the proposed algorithm for CCA is called *truly alternating least-squares (TALS)*. It does not only inherit theoretical properties of global convergence and linear complexity from alternating least-squares but also enjoys a speedup roughly by a factor of $\frac{\sigma_k}{\sigma_k + \sigma_{k+1}}$, where $\sigma_k$ represents the $k$-th largest singular value of $\mathbf{C}$. Most important to practitioners is that remarkable performance improvements can be achieved in practice as will be shown in our experiments, albeit with a slight algorithmic change.

Moreover, we develop another variant of ALS. Inspired by a recent work on accelerated power method [21], we try to think about *faster alternating least-squares (FALS)* for CCA with momentum acceleration. The main idea is to add a momentum term to the update equations of the iterates $\mathbf{\Phi}_t$ and $\mathbf{\Psi}_t$ on top of the truly alternating least-squares, which gives rise to Algorithm 2. Compared to other fast methods, e.g., shift-and-invert preconditioning based methods [20, 1], especially for the block case, the advantage here is that the fast version takes over the simple structure of the plain one and updating iterates remains in a sequential manner. At least, locally linear convergence can be achieved. On the other hand, the algorithm is no longer almost parameter-free due to the momentum parameter which needs to be tuned. Although we can leverage this parameter to pursue better performance by hand-tuning, it requires multiple runs of the algorithm which computationally may not be affordable in practice. To tackle this, we put forward *adaptive alternating least-squares* (AALS) with automatic momentum tuning during iterations, such that it is as easy to use as the plain version and at the same time expected to retain the advantages of the fast one. Experiments show that the adaptive version achieves comparable performance to its predecessor, i.e., the faster alternating least-squares, and often outperforms the truly alternating least-squares.

The rest of the paper is organized as follows. We discuss recent literature in Section 2 and then present our algorithms with convergence guarantees on truly alternating least-squares in Section 3 and the fast versions in Section 4. Our experimental studies are reported in Section 5. Finally, the paper is concluded by discussions in Section 6.

## 2   Related Work

There is a rich literature on CCA. We focus here on the block and off-line algorithms proposed recently. [3] proposed a randomized CCA algorithm for a pair of tall and thin matrices. It first performs a randomized dimensionality reduction on the matrices and then runs an off-the-shelf CCA algorithm for the resulting matrices. However, it seems to have quite a high complexity, and as was pointed out in [16], it does not work for large $d_x$ and $d_y$. To cope with this issue, on top of [3], the problem is cast into solving a sequence of iterative least-squares in [15]. But only sub-optimal results can be achieved this way due to the coarse approximation, which was noted in [10]. [16] proposed an iterative method with a low per-iteration cost, but there is no guarantee of global convergence and the performance is worse than CCALin, i.e., alternating least-squares proposed in [10]. These algorithms directly solve Problem (1).

Alternating least-squares solves Problem (1) indirectly, by targeting an equivalent problem, i.e., generalized eigenspace computation, in the following form:

$$\max_{\mathbf{\Omega}^\top \mathbf{B} \mathbf{\Omega} = \mathbf{I}} \mathrm{tr}(\mathbf{\Omega}^\top \mathbf{A} \mathbf{\Omega}),$$

where $\mathbf{B} = \mathrm{diag}(\mathbf{C}_{xx}, \mathbf{C}_{yy})$. [20] proposed inexact alternating least squares with a sub-linear convergence analysis for the vector case $k = 1$. The block case was considered with block size set to $2k$ and given a linear convergence analysis in [10]. While both algorithms enjoys global convergence, they have the drawbacks mentioned in Section 1. In this paper, our proposed truly alternating least-squares is a natural extension of above two algorithms without the drawbacks.

Most of the fast CCA algorithms rely on the shift-and-invert preconditioning paradigm that is originally designed for eigenvector computation [9]. [20] extended the paradigm to the CCA setting and achieved better performance than alternating least-squares for the vector case. [1] further extended to the block setting, using the vector version as a meta algorithm to recursively find top-$k$ canonical subspaces via deflation. While both algorithms have theoretically faster convergence, pragmatic concerns arise that the shift-and-invert preconditioning paradigm bears a complicated algorithm structure and is difficult to deploy in practice, especially in the block setting. The deployment is built upon a number of tuning parameters including the nontrivial estimation of the spectral gap [20]. The deflation further complicates the task in the block case. In contrast, the fast CCA algorithm presented in this paper follows the momentum acceleration scheme that is also originally designed for eigenvector computation [21], and outperforms alternating least-squares, particularly for the block case. The underlying algorithm is simple with much fewer parameters. Furthermore, the adaptive version does not even need to tune the momentum parameter, making it more practical.

We also note that there are a number of recent CCA algorithms that handle the streaming setting [22, 8, 2, 6, 4]. It will be interesting to investigate how our algorithms extend to this setting.

## 3   Truly Alternating Least-Squares (TALS)

In this section, we detail our proposed *truly alternating least-squares (TALS)* for CCA, starting from the existing alternating least-squares solvers. Update equations of alternating least-squares in [20] can be written as

$$\begin{cases} \tilde{\phi}_{t+1} = \mathbf{C}_{xx}^{-1} \mathbf{C}_{xy} \psi_t + \xi_t, & \phi_{t+1} = \frac{\tilde{\phi}_{t+1}}{\|\tilde{\phi}_{t+1}\|_2} \\ \tilde{\psi}_{t+1} = \mathbf{C}_{yy}^{-1} \mathbf{C}_{xy}^\top \phi_t + \eta_t, & \psi_{t+1} = \frac{\tilde{\psi}_{t+1}}{\|\tilde{\psi}_{t+1}\|_2} \end{cases}, \tag{3}$$

where $\phi_t \in \mathbb{R}^{d_x \times 1}$, $\psi_t \in \mathbb{R}^{d_y \times 1}$, and $\xi_t, \eta_t$ are errors incurred in approximating $\mathbf{C}_{xx}^{-1} \mathbf{C}_{xy} \psi_t$ and $\mathbf{C}_{yy}^{-1} \mathbf{C}_{xy}^\top \phi_t$ by least-squares, respectively. For example, $\mathbf{C}_{xx}^{-1} \mathbf{C}_{xy} \psi_t$ is the exact solution to the linear systems of equations $\mathbf{C}_{xx} \tilde{\phi} = \mathbf{C}_{xy} \psi_t$ with unknowns $\tilde{\phi}$, or equivalently the following least-squares

---
**Algorithm 1** TALS-CCA
---
1: **Input:** $T, k$, data matrices $\mathbf{X}, \mathbf{Y}$.
2: **Output:** approximate top-$k$ canonical subspaces $(\mathbf{\Phi}_T, \mathbf{\Psi}_T)$.
3: Initialize $\mathbf{\Phi}_0 = \mathrm{GS}_{\mathbf{C}_{xx}}(\mathbf{\Phi}_{\mathrm{init}}) \in \mathbb{R}^{d_x \times k}$, $\mathbf{\Psi}_0 = \mathrm{GS}_{\mathbf{C}_{yy}}(\mathbf{\Psi}_{\mathrm{init}}) \in \mathbb{R}^{d_y \times k}$, where entries of
   $\mathbf{\Phi}_{\mathrm{init}}, \mathbf{\Psi}_{\mathrm{init}}$ are i.i.d. standard normal samples.
4: **for** $t = 1, 2, \cdots, T$ **do**
5:   Approximately solve least-squares
$$\tilde{\mathbf{\Phi}}_t \approx \arg \min_{\tilde{\mathbf{\Phi}} \in \mathbb{R}^{d_x \times k}} l_t(\tilde{\mathbf{\Phi}}) = \frac{1}{2n}\|\mathbf{X}^\top \tilde{\mathbf{\Phi}} - \mathbf{Y}^\top \mathbf{\Psi}_{t-1}\|_F^2 + \frac{r_x}{2}\|\tilde{\mathbf{\Phi}}\|_F^2$$
   with initial $\tilde{\mathbf{\Phi}}^{(0)} = \mathbf{\Phi}_{t-1}(\mathbf{\Phi}_{t-1}^\top \mathbf{C}_{xx} \mathbf{\Phi}_{t-1})^{-1}(\mathbf{\Phi}_{t-1}^\top \mathbf{C}_{xy} \mathbf{\Psi}_{t-1})$.
6:   $\mathbf{\Phi}_t = \mathrm{GS}_{\mathbf{C}_{xx}}(\tilde{\mathbf{\Phi}}_t)$.
7:   Approximately solve least-squares
$$\tilde{\mathbf{\Psi}}_t \approx \arg \min_{\tilde{\mathbf{\Psi}} \in \mathbb{R}^{d_y \times k}} s_t(\tilde{\mathbf{\Psi}}) = \frac{1}{2n}\|\mathbf{Y}^\top \tilde{\mathbf{\Psi}} - \mathbf{X}^\top \mathbf{\Phi}_t\|_F^2 + \frac{r_y}{2}\|\tilde{\mathbf{\Psi}}\|_F^2$$
   with initial $\tilde{\mathbf{\Psi}}^{(0)} = \mathbf{\Psi}_{t-1}(\mathbf{\Psi}_{t-1}^\top \mathbf{C}_{yy} \mathbf{\Psi}_{t-1})^{-1}(\mathbf{\Psi}_{t-1}^\top \mathbf{C}_{xy}^\top \mathbf{\Phi}_t)$.
8:   $\mathbf{\Psi}_t = \mathrm{GS}_{\mathbf{C}_{yy}}(\tilde{\mathbf{\Psi}}_t)$.
9: **end for**
---

problem:
$$\min_{\tilde{\phi} \in \mathbb{R}^{d_x \times 1}} l_t(\tilde{\phi}) = \frac{1}{2n}\|\mathbf{X}^\top \tilde{\phi} - \mathbf{Y}^\top \psi_t\| + \frac{r_x}{2}\|\tilde{\phi}\|_2^2.$$

The approximation can be done by running a least-squares solver, warm-started by $\phi_t$, for only a few iterations. The block version for $k > 1$ in [10] needs to take the following form:

$$\begin{cases} \tilde{\mathbf{\Phi}}_{t+1} = \mathbf{C}_{xx}^{-1}\mathbf{C}_{xy}\mathbf{\Psi}_t + \xi_t, \quad \mathbf{\Phi}_{t+1} = \tilde{\mathbf{\Phi}}_{t+1}(\tilde{\mathbf{\Phi}}_{t+1}^\top \mathbf{C}_{xx}\tilde{\mathbf{\Phi}}_{t+1} + \tilde{\mathbf{\Psi}}_{t+1}^\top \mathbf{C}_{yy}\tilde{\mathbf{\Psi}}_{t+1})^{-\frac{1}{2}} \\ \tilde{\mathbf{\Psi}}_{t+1} = \mathbf{C}_{yy}^{-1}\mathbf{C}_{xy}^\top \mathbf{\Phi}_t + \eta_t, \quad \mathbf{\Psi}_{t+1} = \tilde{\mathbf{\Psi}}_{t+1}(\tilde{\mathbf{\Phi}}_{t+1}^\top \mathbf{C}_{xx}\tilde{\mathbf{\Phi}}_{t+1} + \tilde{\mathbf{\Psi}}_{t+1}^\top \mathbf{C}_{yy}\tilde{\mathbf{\Psi}}_{t+1})^{-\frac{1}{2}} \end{cases}, \quad (4)$$

where $\mathbf{\Phi}_t \in \mathbb{R}^{d_x \times 2k}$ and $\mathbf{\Psi}_t \in \mathbb{R}^{d_y \times 2k}$, rather than $\mathbf{\Phi}_t \in \mathbb{R}^{d_x \times k}$ and $\mathbf{\Psi}_t \in \mathbb{R}^{d_y \times k}$. It is easy to see that update equations in both (3) and (4) yield two independent linear systems. It turns out that the independence hampers the empirical performance of alternating least-squares for CCA.

To overcome the drawbacks especially for the block case, we propose the following truly (and inexact) alternating least-squares,

$$\begin{cases} \tilde{\mathbf{\Phi}}_{t+1} = \mathbf{C}_{xx}^{-1}\mathbf{C}_{xy}\mathbf{\Psi}_t + \xi_t, \quad \mathbf{\Phi}_{t+1} = \tilde{\mathbf{\Phi}}_{t+1}(\tilde{\mathbf{\Phi}}_{t+1}^\top \mathbf{C}_{xx}\tilde{\mathbf{\Phi}}_{t+1})^{-\frac{1}{2}} \\ \tilde{\mathbf{\Psi}}_{t+1} = \mathbf{C}_{yy}^{-1}\mathbf{C}_{xy}^\top \mathbf{\Phi}_{t+1} + \eta_{t+1}, \quad \mathbf{\Psi}_{t+1} = \tilde{\mathbf{\Psi}}_{t+1}(\tilde{\mathbf{\Psi}}_{t+1}^\top \mathbf{C}_{yy}\tilde{\mathbf{\Psi}}_{t+1})^{-\frac{1}{2}} \end{cases}, \quad (5)$$

where we now have $\mathbf{\Phi}_t \in \mathbb{R}^{d_x \times k}$ and $\mathbf{\Psi}_t \in \mathbb{R}^{d_y \times k}$. Compared to (3) and (4), two induced linear systems in (5) are coupled together and of half the size in the block setting. Corresponding algorithmic steps are given in Algorithm 1, where subroutine $\mathrm{GS}_{\mathbf{H}}(\cdot)$ performs the generalized Gram-Schmidt orthogonalization process with inner product $\langle,\rangle_{\mathbf{H}}$ for a positive definite matrix $\mathbf{H}$. Note that our initials to the least-squares solver are different from those in both [20] and [10].

Recall that $\mathbf{P}$ and $\mathbf{Q}$ are the top-$k$ left and right singular subspaces of $\mathbf{C}$ with respect to their respective Euclidean metrics, corresponding to singular values $\mathbf{\Sigma} = \mathrm{diag}(\sigma_1, \cdots, \sigma_k)$ in descending order, i.e., $\sigma_i \geq \sigma_j$ for $1 \leq i < j \leq \mathrm{rank}(\mathbf{C})$. Thus, by Equation (2), ground truth $\mathbf{U}$ and $\mathbf{V}$ are the counterparts with respect to metrics $\mathbf{C}_{xx}$ and $\mathbf{C}_{yy}$, respectively. Let $\theta_t = \max\{\theta_{\max}(\mathbf{\Phi}_t, \mathbf{U}), \theta_{\max}(\mathbf{\Psi}_t, \mathbf{V})\}$, where $\theta_{\max}(\mathbf{\Phi}_t, \mathbf{U})$ represents the largest principal angle between subspaces[2] $\mathbf{\Phi}_t$ and $\mathbf{U}$ in underlying metric $\mathbf{C}_{xx}$, i.e., $\theta_{\max}(\mathbf{\Phi}_t, \mathbf{U}) = \cos^{-1}(\sigma_{\min}(\mathbf{U}^\top \mathbf{C}_{xx} \mathbf{\Phi}_t))$. Let $\mathrm{nnz}(\mathbf{X})$ represent the number of nonzero entries in $\mathbf{X}$ and $\kappa(\mathbf{C}_{xx})$ the condition number of $\mathbf{C}_{xx}$. Algorithm 1 then has properties that are described by the following theorem.

**Algorithm 2** FALS-CCA

---

1: **Input:** $T, k$, momentum parameter $\beta$, data matrices $\mathbf{X}, \mathbf{Y}$.
2: **Output:** approximate top-$k$ canonical subspaces $(\mathbf{\Phi}_T, \mathbf{\Psi}_T)$.
3: Initialize $\mathbf{\Phi}_{-1} = \mathbf{0} \in \mathbb{R}^{d_x \times k}$, $\mathbf{\Phi}_0 = \mathrm{GS}_{\mathbf{C}_{xx}}(\mathbf{\Phi}_{\mathrm{init}}) \in \mathbb{R}^{d_x \times k}$, $\mathbf{\Psi}_0 = \mathrm{GS}_{\mathbf{C}_{yy}}(\mathbf{\Psi}_{\mathrm{init}}) \in \mathbb{R}^{d_y \times k}$,
   where entries of $\mathbf{\Phi}_{\mathrm{init}}, \mathbf{\Psi}_{\mathrm{init}}$ are i.i.d. standard normal samples.
4: **for** $t = 1, 2, \cdots, T$ **do**
5:    Approximately solve least-squares
$$\tilde{\mathbf{\Phi}}_t \approx \arg \min_{\tilde{\mathbf{\Phi}} \in \mathbb{R}^{d_x \times k}} l_t(\tilde{\mathbf{\Phi}}) = \frac{1}{2n} \|\mathbf{X}^\top(\tilde{\mathbf{\Phi}} + \beta \mathbf{\Phi}_{t-2}) - \mathbf{Y}^\top \mathbf{\Psi}_{t-1}\|_F^2 + \frac{r_x}{2}\|\tilde{\mathbf{\Phi}} + \beta \mathbf{\Phi}_{t-2}\|_F^2$$
   with initial $\tilde{\mathbf{\Phi}}^{(0)} = \mathbf{\Phi}_{t-1}(\mathbf{\Phi}_{t-1}^\top \mathbf{C}_{xx} \mathbf{\Phi}_{t-1})^{-1}(\mathbf{\Phi}_{t-1}^\top \mathbf{C}_{xy} \mathbf{\Psi}_{t-1})$.
6:    $\mathbf{\Phi}_t = \mathrm{GS}_{\mathbf{C}_{xx}}(\tilde{\mathbf{\Phi}}_t)$.
7:    Approximately solve least-squares
$$\tilde{\mathbf{\Psi}}_t \approx \arg \min_{\tilde{\mathbf{\Psi}} \in \mathbb{R}^{d_y \times k}} s_t(\tilde{\mathbf{\Psi}}) = \frac{1}{2n} \|\mathbf{Y}^\top(\tilde{\mathbf{\Psi}} + \beta \mathbf{\Psi}_{t-1}) - \mathbf{X}^\top \mathbf{\Phi}_t\|_F^2 + \frac{r_y}{2}\|\tilde{\mathbf{\Psi}} + \beta \mathbf{\Psi}_{t-1}\|_F^2$$
   with initial $\tilde{\mathbf{\Psi}}^{(0)} = \mathbf{\Psi}_{t-1}(\mathbf{\Psi}_{t-1}^\top \mathbf{C}_{yy} \mathbf{\Psi}_{t-1})^{-1}(\mathbf{\Psi}_{t-1}^\top \mathbf{C}_{xy}^\top \mathbf{\Phi}_t)$.
8:    $\mathbf{\Psi}_t = \mathrm{GS}_{\mathbf{C}_{yy}}(\tilde{\mathbf{\Psi}}_t)$.
9: **end for**

---

**Theorem 1** *Given data matrices* $\mathbf{X}$ *and* $\mathbf{Y}$, *TALS-CCA computes a* $d_x \times k$ *matrix* $\mathbf{\Phi}_T$ *and a* $d_y \times k$ *matrix* $\mathbf{\Psi}_T$ *which are estimates of top-$k$ canonical subspaces* $(\mathbf{U}, \mathbf{V})$ *with an error of* $\epsilon$, *i.e.,* $\mathbf{\Phi}_T^\top \mathbf{C}_{xx} \mathbf{\Phi}_T = \mathbf{\Psi}_T^\top \mathbf{C}_{yy} \mathbf{\Psi}_T = \mathbf{I}$ *and* $\tan \theta_T \leq \epsilon$, *in* $T = O(\frac{\sigma_k^2}{\sigma_k^2 - \sigma_{k+1}^2} \log \frac{1}{\epsilon \cos \theta_0})$ *iterations. If Nesterov's accelerated gradient descent is used as the least-squares solver, the running time is at most*

$$O(\frac{k\sigma_k^2}{\sigma_k^2 - \sigma_{k+1}^2} \mathrm{nnz}(\mathbf{X}, \mathbf{Y})\kappa(\mathbf{X}, \mathbf{Y})(\log \frac{1}{\cos \theta_0} \log \frac{\sigma_1}{(\sigma_k^2 - \sigma_{k+1}^2)\cos \theta_0} +$$
$$\log \frac{1}{\epsilon} \log \frac{\sigma_1}{\sigma_k^2 - \sigma_{k+1}^2}) + \frac{k^2 \sigma_k^2}{\sigma_k^2 - \sigma_{k+1}^2} \max\{d_x, d_y\} \log \frac{1}{\epsilon \cos \theta_0}),$$

*where* $\mathrm{nnz}(\mathbf{X}, \mathbf{Y}) = \mathrm{nnz}(\mathbf{X}) + \mathrm{nnz}(\mathbf{Y})$ *and* $\kappa(\mathbf{X}, \mathbf{Y}) = \max\{\kappa(\mathbf{C}_{xx}), \kappa(\mathbf{C}_{yy})\}$.

Note that random initializations to $\mathbf{\Phi}_0$ and $\mathbf{\Psi}_0$ result in $\cos \theta_0 > 0$ with high probability, by Lemma 13 in [10]. Thus, TALS-CCA is globally and linearly convergent. Proofs are provided in the supplementary material. Compared to alternating least-squares, e.g., CCALin in [10], it is roughly faster by a factor of $\frac{\sigma_k}{\sigma_k + \sigma_{k+1}}$, whereas empirical improvements are often more pronounced. Note that it makes a difference especially for the cases of a small singular value gap $\sigma_k - \sigma_{k+1}$.

## 4 Faster Alternating Least-Squares (FALS)

In this section, we consider the momentum acceleration for CCA, motivated by accelerated power method for eigenvector computation [21]. To derive update equations for faster alternating least-squares (FALS), we first have CCA cast into the setting of eigenvector computation on real symmetric matrices and then introduce the momentum to speedup.

Recall that

$$\mathbf{A} = \begin{pmatrix} & \mathbf{C}_{xy} \\ \mathbf{C}_{xy}^\top & \end{pmatrix} \quad \text{and} \quad \mathbf{B} = \begin{pmatrix} \mathbf{C}_{xx} & \\ & \mathbf{C}_{yy} \end{pmatrix}.$$

Let

$$\tilde{\mathbf{W}}_t = \mathbf{B}^{\frac{1}{2}} \begin{pmatrix} \tilde{\mathbf{\Phi}}_t \\ \tilde{\mathbf{\Psi}}_t \end{pmatrix} \in \mathbb{R}^{(d_x + d_y) \times 2k} \quad \text{and} \quad \mathbf{W}_t = \mathbf{B}^{\frac{1}{2}} \begin{pmatrix} \mathbf{\Phi}_t \\ \mathbf{\Psi}_t \end{pmatrix} \in \mathbb{R}^{(d_x + d_y) \times 2k}.$$

The momentum acceleration applied to $\mathbf{B}^{-\frac{1}{2}} \mathbf{A} \mathbf{B}^{-\frac{1}{2}}$ then yields that

$$\tilde{\mathbf{W}}_{t+1} = \mathbf{B}^{-\frac{1}{2}} \mathbf{A} \mathbf{B}^{-\frac{1}{2}} \mathbf{W}_t - \beta \mathbf{W}_{t-1}, \quad \mathbf{W}_{t+1} = \tilde{\mathbf{W}}_{t+1}(\tilde{\mathbf{W}}_{t+1}^\top \tilde{\mathbf{W}}_{t+1})^{-\frac{1}{2}}, \tag{6}$$

---

**Algorithm 3** AALS-CCA

1: **Input:** $T, k$, data matrices $\mathbf{X}, \mathbf{Y}$.
2: **Output:** approximate top-$k$ canonical subspaces $(\mathbf{\Phi}_T, \mathbf{\Psi}_T)$.
3: Initialize $\mathbf{\Phi}_{-1} = \mathbf{0} \in \mathbb{R}^{d_x \times k}$, $\mathbf{\Phi}_0 = \mathrm{GS}_{\mathbf{C}_{xx}}(\mathbf{\Phi}_{\mathrm{init}}) \in \mathbb{R}^{d_x \times k}$, $\mathbf{\Psi}_0 = \mathrm{GS}_{\mathbf{C}_{yy}}(\mathbf{\Psi}_{\mathrm{init}}) \in \mathbb{R}^{d_y \times k}$,
   where entries of $\mathbf{\Phi}_{\mathrm{init}}, \mathbf{\Psi}_{\mathrm{init}}$ are i.i.d. standard normal samples.
4: **for** $t = 1, 2, \cdots, T$ **do**
5:      Set $\beta_{t,1} = \dfrac{1}{4} \min\limits_{1 \le i \le k} (\mathbf{\Sigma}_{ii}^{(t-1,1)})^2$, where $\mathbf{\Sigma}^{(t-1,1)} = (\mathbf{\Phi}_{t-1}^\top \mathbf{C}_{xx} \mathbf{\Phi}_{t-1})^{-1} (\mathbf{\Phi}_{t-1}^\top \mathbf{C}_{xy} \mathbf{\Psi}_{t-1})$.
6:      Approximately solve least-squares
$$\tilde{\mathbf{\Phi}}_t \approx \arg \min_{\tilde{\mathbf{\Phi}} \in \mathbb{R}^{d_x \times k}} l_t(\tilde{\mathbf{\Phi}}) = \frac{1}{2n} \|\mathbf{X}^\top (\tilde{\mathbf{\Phi}} + \beta_{t,1} \mathbf{\Phi}_{t-2}) - \mathbf{Y}^\top \mathbf{\Psi}_{t-1}\|_F^2 + \frac{r_x}{2} \|\tilde{\mathbf{\Phi}} + \beta_{t,1} \mathbf{\Phi}_{t-2}\|_F^2$$
     with initial $\tilde{\mathbf{\Phi}}^{(0)} = \mathbf{\Phi}_{t-1} \mathbf{\Sigma}^{(t-1,1)}$.
7:      $\mathbf{\Phi}_t = \mathrm{GS}_{\mathbf{C}_{xx}}(\tilde{\mathbf{\Phi}}_t)$.
8:      Set $\beta_{t,2} = \dfrac{1}{4} \min\limits_{1 \le i \le k} (\mathbf{\Sigma}_{ii}^{(t-1,2)})^2$, where $\mathbf{\Sigma}^{(t-1,2)} = (\mathbf{\Psi}_{t-1}^\top \mathbf{C}_{yy} \mathbf{\Psi}_{t-1})^{-1} (\mathbf{\Psi}_{t-1}^\top \mathbf{C}_{xy}^\top \mathbf{\Phi}_t)$.
9:      Approximately solve least-squares
$$\tilde{\mathbf{\Psi}}_t \approx \arg \min_{\tilde{\mathbf{\Psi}} \in \mathbb{R}^{d_y \times k}} s_t(\tilde{\mathbf{\Psi}}) = \frac{1}{2n} \|\mathbf{Y}^\top (\tilde{\mathbf{\Psi}} + \beta_{t,2} \mathbf{\Psi}_{t-1}) - \mathbf{X}^\top \mathbf{\Phi}_t\|_F^2 + \frac{r_y}{2} \|\tilde{\mathbf{\Psi}} + \beta_{t,2} \mathbf{\Psi}_{t-1}\|_F^2$$
     with initial $\tilde{\mathbf{\Psi}}^{(0)} = \mathbf{\Psi}_{t-1} \mathbf{\Sigma}^{(t-1,2)}$.
10:     $\mathbf{\Psi}_t = \mathrm{GS}_{\mathbf{C}_{yy}}(\tilde{\mathbf{\Psi}}_t)$.
11: **end for**

---

where $-\beta \mathbf{W}_{t-1}$ is known as the momentum term and $\beta$ is the momentum parameter. Expanding the above update equation into two inexact update equations in $\mathbf{\Phi}_t$, $\mathbf{\Psi}_t$ and having them coupled together as with TALS, we arrive at our faster (truly and inexact) alternating least-squares as follows:

$$\begin{cases} \tilde{\mathbf{\Phi}}_{t+1} = \mathbf{C}_{xx}^{-1} \mathbf{C}_{xy} \mathbf{\Psi}_t - \beta \mathbf{\Phi}_{t-1} + \xi_t, & \mathbf{\Phi}_{t+1} = \tilde{\mathbf{\Phi}}_{t+1} (\tilde{\mathbf{\Phi}}_{t+1}^\top \mathbf{C}_{xx} \tilde{\mathbf{\Phi}}_{t+1})^{-\frac{1}{2}} \\ \tilde{\mathbf{\Psi}}_{t+1} = \mathbf{C}_{yy}^{-1} \mathbf{C}_{xy}^\top \mathbf{\Phi}_{t+1} - \beta \mathbf{\Psi}_t + \eta_{t+1}, & \mathbf{\Psi}_{t+1} = \tilde{\mathbf{\Psi}}_{t+1} (\tilde{\mathbf{\Psi}}_{t+1}^\top \mathbf{C}_{yy} \tilde{\mathbf{\Psi}}_{t+1})^{-\frac{1}{2}} \end{cases},$$

where $\mathbf{\Psi}_t \in \mathbb{R}^{d_x \times k}$ and $\mathbf{\Phi}_t \in \mathbb{R}^{d_y \times k}$. The algorithmic steps are given in Algorithm 2 which keeps as simple as the plain alternating least-squares. Despite the simplicity, the analysis of faster convergence is very difficult (see our discussions in Section 6). Nonetheless, it is at least locally linearly convergent, as stated in the following theorem.

**Theorem 2** *Given data matrices $\mathbf{X}$ and $\mathbf{Y}$, FALS-CCA computes a $d_x \times k$ matrix $\mathbf{\Phi}_T$ and a $d_y \times k$ matrix $\mathbf{\Psi}_T$ which are estimates of top-$k$ canonical subspaces $(\mathbf{U}, \mathbf{V})$ with an error of $\epsilon$, i.e., $\mathbf{\Phi}_T^\top \mathbf{C}_{xx} \mathbf{\Phi}_T = \mathbf{\Psi}_T^\top \mathbf{C}_{yy} \mathbf{\Psi}_T = \mathbf{I}$ and $\tan \theta_T \le \epsilon$, in $T = O(\frac{\sigma_k^2 - c\sigma_1 \beta}{\sigma_k^2 - \sigma_{k+1}^2 - 4c\sigma_1 \beta} \log \frac{1}{\epsilon \cos \theta_0})$ iterations if $\theta_0 \le \frac{\pi}{4}$. If Nesterov's accelerated gradient descent is used as the least-squares solver, the running time is at most*

$$O(\frac{k(\sigma_k^2 - c\sigma_1 \beta)}{\sigma_k^2 - \sigma_{k+1}^2 - 4c\sigma_1 \beta} \mathrm{nnz}(\mathbf{X}, \mathbf{Y}) \kappa(\mathbf{X}, \mathbf{Y}) (\log \frac{1}{\cos \theta_0} \log \frac{\sigma_1}{(\sigma_k^2 - \sigma_{k+1}^2) \cos \theta_0} +$$

$$\log \frac{1}{\epsilon} \log \frac{\sigma_1}{\sigma_k^2 - \sigma_{k+1}^2}) + \frac{k^2(\sigma_k^2 - c\sigma_1 \beta)}{\sigma_k^2 - \sigma_{k+1}^2 - 4c\sigma_1 \beta} \max\{d_x, d_y\} \log \frac{1}{\epsilon \cos \theta_0}),$$

*where $c > 0$ is a constant.*

Clearly, the momentum parameter plays a key role for Algorithm 2 to work. It is central for us to figure out sensible ways to set it in practice. Given the tight analysis (see Theorem 11 in [21]) for the exact update (6) in $\mathbf{W}_t$, the optimal value of $\beta$ should be around $\frac{\sigma_{k+1}^2}{4}$. On the other hand, it holds for the optimal solution that

$$\mathbf{C}_{xy} \mathbf{V} = \mathbf{C}_{xx} \mathbf{U} \mathbf{\Sigma}, \quad \mathbf{C}_{xy}^\top \mathbf{U} = \mathbf{C}_{yy} \mathbf{V} \mathbf{\Sigma}.$$

We thus can write that

$$(\mathbf{U}^\top \mathbf{C}_{xx} \mathbf{U})^{-1} \mathbf{U}^\top \mathbf{C}_{xy} \mathbf{V} = \mathbf{\Sigma} = (\mathbf{V}^\top \mathbf{C}_{yy} \mathbf{V})^{-1} \mathbf{V}^\top \mathbf{C}_{xy}^\top \mathbf{U}.$$

Accordingly, we have the following estimate options of $\mathbf{\Sigma}$ for sufficiently large $t$:

$$\begin{aligned}
\mathbf{\Sigma}^{(t,1)} &\triangleq (\mathbf{\Phi}_t^\top \mathbf{C}_{xx} \mathbf{\Phi}_t)^{-1} (\mathbf{\Phi}_t^\top \mathbf{C}_{xy} \mathbf{\Psi}_t), \\
\mathbf{\Sigma}^{(t,2)} &\triangleq (\mathbf{\Psi}_t^\top \mathbf{C}_{yy} \mathbf{\Psi}_t)^{-1} (\mathbf{\Psi}_t^\top \mathbf{C}_{xy}^\top \mathbf{\Phi}_{t+1}), \\
\mathbf{\Sigma}^{(t,3)} &\triangleq (\mathbf{\Phi}_t^\top \mathbf{C}_{xx} \mathbf{\Phi}_t + \mathbf{\Psi}_t^\top \mathbf{C}_{yy} \mathbf{V})^{-1} (\mathbf{\Phi}_t^\top \mathbf{C}_{xy} \mathbf{\Psi}_t + \mathbf{\Psi}_t^\top \mathbf{C}_{xy}^\top \mathbf{\Phi}_t).
\end{aligned}$$

Before iterates $\mathbf{\Phi}_t$ and $\mathbf{\Psi}_t$ converge, $\min_{1 \leq i \leq k} \mathbf{\Sigma}_{ii}^{(t,j)}$ is strictly less than $\sigma_k$ in general. Meanwhile, $\sigma_{k+1}$ is bounded above by $\sigma_k$. Therefore, our first strategy for approximating the optimal momentum parameter is to run a small number of iterations of TALS, which can be viewed as a burning process, and then set $\beta_j = \frac{1}{4} \min_{1 \leq i \leq k} (\mathbf{\Sigma}_{ii}^{(T_0,j)})^2$ for FALS. Denote the resulting algorithm as FALS-$T_0$.

**Adaptive Alternating Least-Squares** (AALS)   To further avoid choosing burning parameter $T_0$, the second strategy we propose is to automatically and adaptively adjust momentum parameter $\beta$ during iterations, as described in Algorithm 3. Compared to Algorithm 2, there is no additional cost in running AALS. It keeps as easy to use as the plain alternating least-squares while retaining the advantages of the fast version.

## 5   Experiments

In this section, we examine and compare the empirical behaviors of both existing and our alternating least-squares algorithms. Three real-world datasets are used: Mediamill [18], JW11 [17], and MNIST [14]. See Table 1 for the statistics and descriptions. They are commonly used to test CCA

Table 1: Statistics of Datasets

| DATA | Description | $d_x$ | $d_y$ | $n$ |
|------|-------------|-------|-------|-----|
| Memdiamill | images and its labels | 100 | 120 | 30000 |
| JW11 | acoustic and articulation measurements | 273 | 112 | 30000 |
| MNIST | left and right halves of images | 392 | 392 | 60000 |
| Youtube | UCI Youtube audio and vision streams | 64 | 1024 | 122041 |

solvers [20]. In order to show the inability of the plain alternating least-squares with block size $k$ to solve CCA, we adapt alternating least-squares in both [20] and [10] to block size $k$, denoted as ALS-$k$ and CCALin-$k$, respectively. Note that the post-processing step is not needed any more for CCALin-$k$. Two algorithms differ only in the initial to the least-squares solver. The original CCALin algorithm is also included as a baseline. They are compared with our TALS, FALS-$T_0$ (i.e., FALS with burning parameter $T_0$), and AALS. Particularly, $T_0 \in \{4, 6\}$ is used. Regularization parameters are fixed to $r_x = r_y = 0.1$. Stochastic variance reduced gradient (SVRG) is the least-squares solver we use for each algorithm. Throughout the experiments the solver runs 2 epochs with each running $n$ iterations with constant step-sizes $\alpha_{\mathbf{\Phi}} = \frac{1}{\max_i \|\mathbf{x}_i\|_2^2}$ for $\mathbf{\Phi}_t$ and $\alpha_{\mathbf{\Psi}} = \frac{1}{\max_i \|\mathbf{y}_i\|_2^2}$ for $\mathbf{\Psi}_t$, where $\mathbf{x}_i$ is the $i$-th column of $\mathbf{X}$. All the algorithms were implemented in MATLAB, and run on a laptop with 8 GB memory. Quality measures we use are as follows:

- $\sin^2 \theta_u \triangleq \sin^2 \theta_{\max}(\mathbf{\Phi}_t, \mathbf{U})$, squared sine value of largest principal angle between $\mathbf{\Phi}_t$ and $\mathbf{U}$;

- $\sin^2 \theta_v \triangleq \sin^2 \theta_{\max}(\mathbf{\Psi}_t, \mathbf{V})$, squared sine value of largest principal angle between $\mathbf{\Psi}_t$ and $\mathbf{V}$,

where ground truth $(\mathbf{P}, \mathbf{\Sigma}, \mathbf{Q})$ is obtained by MATLAB's svds function for evaluation purpose. Smaller is better for each measure. It is worth noting that the two measures are more indicative of the performance of all the algorithms considered here, compared to the relative objective function error measure

$$\frac{f^\star - f}{f^\star} \triangleq \frac{\mathrm{tr}(\mathbf{\Sigma}) - \mathrm{tr}(\mathbf{\Phi}_t^\top \mathbf{C}_{xy} \mathbf{\Psi}_t)}{\mathrm{tr}(\mathbf{\Sigma})},$$

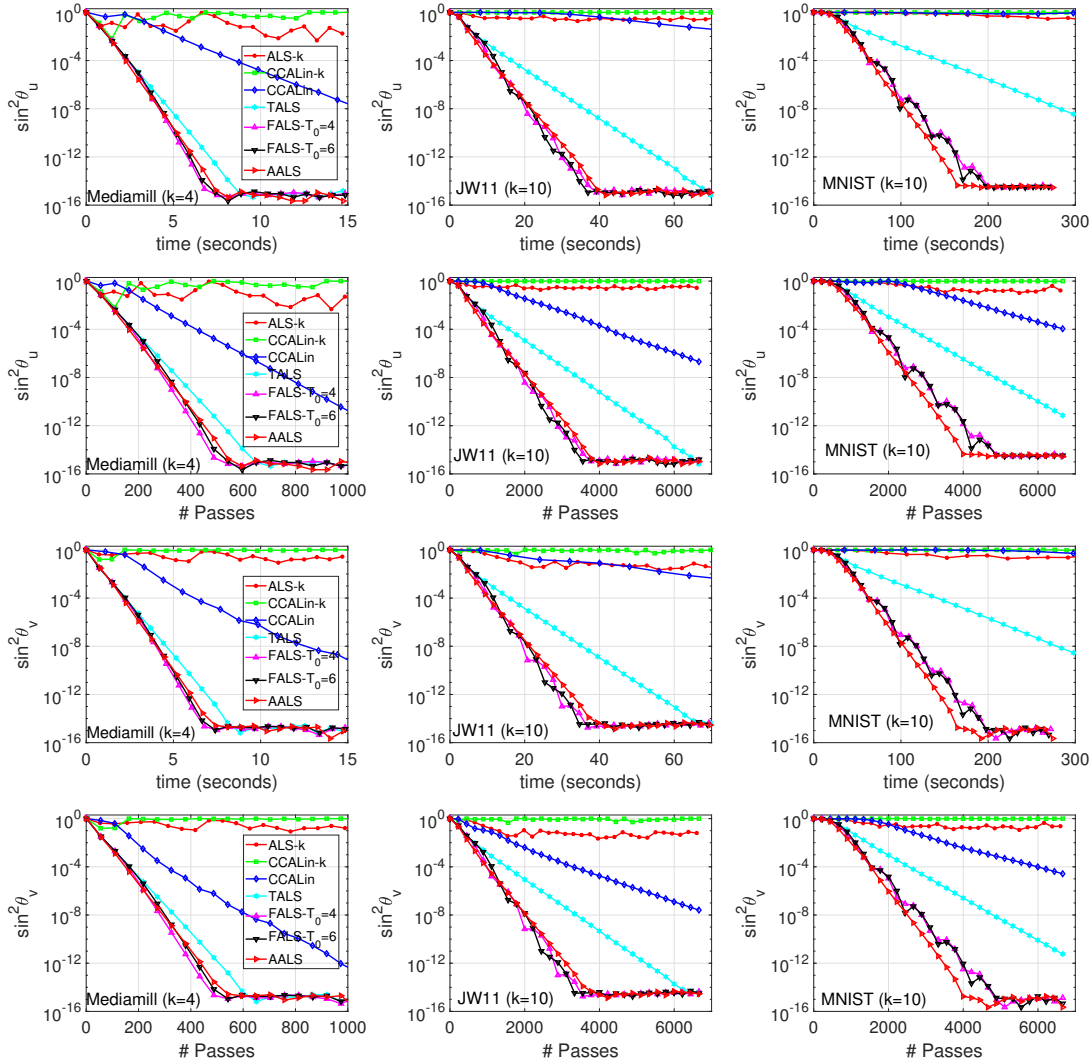

Figure 1: Performance of different ALS algorithms for CCA.

because they do not directly optimize the objective function of Problem (1), i.e., $\mathrm{tr}(\boldsymbol{\Phi}^\top \mathbf{C}_{xy} \boldsymbol{\Psi})$, especially for the CCALin. Convergence results in terms of $(f^\star - f)/f^\star$ are reported in the supplementary material.

Convergence curves of all the considered ALS algorithms are plotted in a $4 \times 3$ array of figures in Figure 1 with a column for each dataset. Upper and lower halves of the rows of figures correspond to $\sin^2 \theta_u$ and $\sin^2 \theta_v$, respectively, while upper and lower rows in each half correspond to results in running time and passes over data, respectively. Note that the curve patterns in running time and passes are not necessarily the same, e.g., for CCALin. From these empirical results, we first observe that both ALS-$k$ and CCALin-$k$ indeed fail to work as the values of both measures always remain high during iterations across datasets. This is because the target ground-truth of both algorithms does not cover top-$k$ canonical subspaces. Second, it takes a much longer time for the CCALin than our ALS algorithms to find a solution even with low precision. Third, our TALS achieves better performance than the CCALin by a large margin in both measures, demonstrating the advantage of the coupling in ALS for CCA. Last, further speedups over TALS are observed for the fast versions, which showcases the potential of the momentum acceleration for CCA. Particularly, the adaptive version, i.e., AALS, without the need to tune the momentum parameter and set the burning parameter, performs equally well as FALS-$T_0$, proving its practical value to certain extent.

Additional experiments are provided in the supplementary material, aiming to demonstrate: 1) the performance of all the considered algorithms with varying block sizes; 2) the performance of our ALS algorithms especially the fast versions in comparison with the shift-and-invert (SI) preconditioning method [20] in the vector setting; 3) the performance of the algorithms on more datasets ($n = 122041$). These experiments indicate that the truly alternating least-squares sometimes can achieve equally good performance compared to its fast versions. In the vector case, the faster alternating least-squares may even work better than the SI method which, though is given the advantage of the knowledge on the spectral gap at $k = 1$ and other tuning parameters.

## 6    Discussion

In this work, we study alternating least-squares as a block CCA solver. Noting the drawbacks of current alternating least-squares methods, we propose the truly alternating least-squares which only needs to update equations of half the size due to coupling. Both theory and practice show that the coupling can significantly improve the performance of alternating least-squares. On top of that, we further propose faster alternating least-squares with momentum acceleration. To make it practical, two strategies are put forward to set the momentum parameter. One is to introduce a burning phase to set it by running the truly alternating least-squares for a few iterations. The other is to automatically adjust the momentum parameter during iterations, making it as easy to use as the plain alternating least-squares without sacrificing fast convergence. Experiments show that both strategies work well. Despite the excellent performance of the fast versions, it lacks of a tight convergence analysis explaining the empirical behaviors. This seems quite difficult, given that there has been no such theory thus far on the momentum acceleration for the basic eigenvector computation in a corresponding setting. The coupling in our context further complicates the analysis. We leave it to our future work where other settings, e.g., streaming or robust, may be considered as well.

## Footnotes

[1]We assume that $\mathbf{X}$ and $\mathbf{Y}$ are row centered at the origin.

[2]For brevity we use $\mathbf{\Phi}_t$ to represent the subspace spanned by columns of $\mathbf{\Phi}_t$ or one of its bases in the underlying metric $\mathbf{C}_{xx}$ without any risk of confusion.

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
