[Supplementary Material]

# Supplementary Material: Towards Practical Alternating Least-Squares for CCA

**Zhiqiang Xu and Ping Li**

Cognitive Computing Lab
Baidu Research
No.10 Xibeiwang East Road, Beijing, 10085, China
10900 NE 8th St, Bellevue, WA 98004, USA
{xuzhiqiang04,liping11}@baidu.com

**Notations**   Assume that $r = r(\mathbf{C})$, where $r(\cdot)$ represents the rank of a matrix. Let $(\mathbf{u}_i, \mathbf{v}_i, \sigma_i)$ be the $i$-th largest singular value triplet of $\mathbf{C}$, $1 \le i \le r \le \min\{d_x, d_y\}$, where $\sigma_i$ represents the $i$-th largest singular value and $\mathbf{u}_i, \mathbf{v}_i$ represents the corresponding left and right singular vectors of unit length in the sense that $\mathbf{u}_i^\top \mathbf{C}_{xx} \mathbf{u}_i = \mathbf{v}_i^\top \mathbf{C}_{yy} \mathbf{v}_i = 1$, respectively. Denote for $1 \le k \le r$ that

$$\mathbf{U} = (\mathbf{u}_1, \cdots, \mathbf{u}_k), \quad \boldsymbol{\Sigma} = \mathrm{diag}(\sigma_1, \cdots, \sigma_k), \quad \mathbf{V} = (\mathbf{v}_1, \cdots, \mathbf{v}_k),$$
$$\mathbf{U}_\perp = (\mathbf{u}_{k+1}, \cdots, \mathbf{u}_r), \boldsymbol{\Sigma}_\perp = \mathrm{diag}(\sigma_{k+1}, \cdots, \sigma_r), \mathbf{V}_\perp = (\mathbf{v}_{k+1}, \cdots, \mathbf{v}_r).$$

We then have that

$$\mathbf{C}_{xy} = \mathbf{C}_{xx}(\mathbf{U}\boldsymbol{\Sigma}\mathbf{V}^\top + \mathbf{U}_\perp \boldsymbol{\Sigma}_\perp \mathbf{V}_\perp^\top)\mathbf{C}_{yy}, \tag{1}$$

where

$$\mathbf{U}^\top \mathbf{C}_{xx} \mathbf{U} = \mathbf{I}, \qquad \mathbf{U}_\perp^\top \mathbf{C}_{xx} \mathbf{U}_\perp = \mathbf{I},$$
$$\mathbf{V}^\top \mathbf{C}_{yy} \mathbf{V} = \mathbf{I}, \qquad \mathbf{V}_\perp^\top \mathbf{C}_{yy} \mathbf{V}_\perp = \mathbf{I},$$
$$\mathbf{U}^\top \mathbf{C}_{xx} \mathbf{U}_\perp = \mathbf{0}, \quad \mathbf{V}^\top \mathbf{C}_{yy} \mathbf{V}_\perp = \mathbf{0}.$$

**Theorem 1**   Given data matrices $\mathbf{X}$ and $\mathbf{Y}$, TALS-CCA computes a $d_x \times k$ matrix $\boldsymbol{\Phi}_T$ and a $d_y \times k$ matrix $\boldsymbol{\Psi}_T$ which are estimates of top-$k$ canonical subspaces $(\mathbf{U}, \mathbf{V})$ with an error of $\epsilon$, i.e., $\boldsymbol{\Phi}_T^\top \mathbf{C}_{xx} \boldsymbol{\Phi}_T = \boldsymbol{\Psi}_T^\top \mathbf{C}_{yy} \boldsymbol{\Psi}_T = \mathbf{I}$ and $\tan\theta_T \le \epsilon$, in $T = O(\frac{\sigma_k^2}{\sigma_k^2 - \sigma_{k+1}^2} \log \frac{1}{\epsilon \cos\theta_0})$ iterations. If Nesterov's accelerated gradient descent is used as the least-squares solver, the running time is at most

$$O(\frac{k\sigma_k^2}{\sigma_k^2 - \sigma_{k+1}^2} \mathrm{nnz}(\mathbf{X}, \mathbf{Y})\kappa(\mathbf{X}, \mathbf{Y})(\log\frac{1}{\cos\theta_0} \log \frac{\sigma_1}{(\sigma_k^2 - \sigma_{k+1}^2)\cos\theta_0} +$$
$$\log\frac{1}{\epsilon} \log \frac{\sigma_1}{\sigma_k^2 - \sigma_{k+1}^2}) + \frac{k^2\sigma_k^2}{\sigma_k^2 - \sigma_{k+1}^2} \max\{d_x, d_y\} \log \frac{1}{\epsilon\cos\theta_0}),$$

where $\mathrm{nnz}(\mathbf{X}, \mathbf{Y}) = \mathrm{nnz}(\mathbf{X}) + \mathrm{nnz}(\mathbf{Y})$ and $\kappa(\mathbf{X}, \mathbf{Y}) = \max\{\kappa(\mathbf{C}_{xx}), \kappa(\mathbf{C}_{yy})\}$.

**Proof**   We follow the analysis of [1] closely. To analyze $\tan\theta_{t+1}$, we focus on $\tan\theta_{\max}(\boldsymbol{\Phi}_{t+1}, \mathbf{U})$ and the case of $\tan\theta_{\max}(\boldsymbol{\Psi}_{t+1}, \mathbf{V})$ is analogous. The coupled and inexact update equations of our TALS are as follows,

$$\begin{cases} \tilde{\boldsymbol{\Phi}}_{t+1} = \mathbf{C}_{xx}^{-1}\mathbf{C}_{xy}\boldsymbol{\Psi}_t + \xi_t, & \boldsymbol{\Phi}_{t+1} = \tilde{\boldsymbol{\Phi}}_{t+1}\mathbf{R}_{t+1} \\ \tilde{\boldsymbol{\Psi}}_{t+1} = \mathbf{C}_{yy}^{-1}\mathbf{C}_{xy}^\top \boldsymbol{\Phi}_{t+1} + \eta_{t+1}, & \boldsymbol{\Psi}_{t+1} = \tilde{\boldsymbol{\Psi}}_{t+1}\mathbf{S}_{t+1} \end{cases},$$

where $\mathbf{R}_{t+1} = (\tilde{\boldsymbol{\Phi}}_{t+1}^\top \mathbf{C}_{xx} \tilde{\boldsymbol{\Phi}}_{t+1})^{-\frac{1}{2}}$ and $\mathbf{S}_{t+1} = (\tilde{\boldsymbol{\Psi}}_{t+1}^\top \mathbf{C}_{yy} \tilde{\boldsymbol{\Psi}}_{t+1})^{-\frac{1}{2}}$. Particularly, we assume that

$$\max\{\|\xi_t\|_{\mathbf{C}_{xx}, F}, \|\eta_t\|_{\mathbf{C}_{yy}, F}\} \le \frac{\sigma_k^2 - \sigma_{k+1}^2}{12} \min\{\sin\theta_t, \cos\theta_t\}. \tag{2}$$

By Lemma 1, we then have that
$$\tan\theta_{\max}(\boldsymbol{\Phi}_{t+1},\mathbf{U}) = \|\mathbf{U}_\perp^\top\mathbf{C}_{xx}\boldsymbol{\Phi}_{t+1}(\mathbf{U}^\top\mathbf{C}_{xx}\boldsymbol{\Phi}_{t+1})^{-1}\|_2.$$
To proceed, note that
$$\boldsymbol{\Phi}_{t+1} = (\mathbf{C}_{xx}^{-1}\mathbf{C}_{xy}(\mathbf{C}_{yy}^{-1}\mathbf{C}_{xy}^\top\boldsymbol{\Phi}_t + \eta_t)\mathbf{S}_t + \xi_t)\mathbf{R}_{t+1}. \qquad (3)$$
Using Eq. (1), we have
$$\begin{aligned}
\mathbf{C}_{xy}\mathbf{C}_{yy}^{-1}\mathbf{C}_{xy}^\top &= \mathbf{C}_{xx}(\mathbf{U}\boldsymbol{\Sigma}\mathbf{V}^\top + \mathbf{U}_\perp\boldsymbol{\Sigma}_\perp\mathbf{V}_\perp^\top)\mathbf{C}_{yy}(\mathbf{V}\boldsymbol{\Sigma}\mathbf{U}^\top + \mathbf{V}_\perp\boldsymbol{\Sigma}_\perp\mathbf{U}_\perp^\top)\mathbf{C}_{xx}\\
&= \mathbf{C}_{xx}(\mathbf{U}\boldsymbol{\Sigma}^2\mathbf{U}^\top + \mathbf{U}_\perp\boldsymbol{\Sigma}_\perp^2\mathbf{U}_\perp^\top)\mathbf{C}_{xx}.
\end{aligned}$$
Accordingly, one gets that
$$\begin{aligned}
\mathbf{U}_\perp^\top\mathbf{C}_{xx}\boldsymbol{\Phi}_{t+1} &= (\boldsymbol{\Sigma}_\perp^2\mathbf{U}_\perp^\top\mathbf{C}_{xx}\boldsymbol{\Phi}_t\mathbf{S}_t + \mathbf{U}_\perp^\top\mathbf{C}_{xy}\eta_t\mathbf{S}_t + \mathbf{U}_\perp^\top\mathbf{C}_{xx}\xi_t)\mathbf{R}_{t+1},\\
\mathbf{U}^\top\mathbf{C}_{xx}\boldsymbol{\Phi}_{t+1} &= (\boldsymbol{\Sigma}^2\mathbf{U}^\top\mathbf{C}_{xx}\boldsymbol{\Phi}_t\mathbf{S}_t + \mathbf{U}^\top\mathbf{C}_{xy}\eta_t\mathbf{S}_t + \mathbf{U}^\top\mathbf{C}_{xx}\xi_t)\mathbf{R}_{t+1}.
\end{aligned}$$
Thus, we can write that
$$\tan\theta_{\max}(\boldsymbol{\Phi}_{t+1},\mathbf{U})$$
$$= \|(\boldsymbol{\Sigma}_\perp^2\mathbf{U}_\perp^\top\mathbf{C}_{xx}\boldsymbol{\Phi}_t\mathbf{S}_t + \mathbf{U}_\perp^\top\mathbf{C}_{xy}\eta_t\mathbf{S}_t + \mathbf{U}_\perp^\top\mathbf{C}_{xx}\xi_t) \cdot$$
$$\quad (\boldsymbol{\Sigma}^2\mathbf{U}^\top\mathbf{C}_{xx}\boldsymbol{\Phi}_t\mathbf{S}_t + \mathbf{U}^\top\mathbf{C}_{xy}\eta_t\mathbf{S}_t + \mathbf{U}^\top\mathbf{C}_{xx}\xi_t)^{-1}\|_2$$
$$\leq \frac{\|(\boldsymbol{\Sigma}_\perp^2\mathbf{U}_\perp^\top\mathbf{C}_{xx}\boldsymbol{\Phi}_t\mathbf{S}_t + \mathbf{U}_\perp^\top\mathbf{C}_{xy}\eta_t\mathbf{S}_t + \mathbf{U}_\perp^\top\mathbf{C}_{xx}\xi_t)(\mathbf{U}^\top\mathbf{C}_{xx}\boldsymbol{\Phi}_t\mathbf{S}_t)^{-1}\|_2}{\sigma_{\min}(\boldsymbol{\Sigma}^2 + \mathbf{U}^\top\mathbf{C}_{xy}\eta_t\mathbf{S}_t(\mathbf{U}^\top\mathbf{C}_{xx}\boldsymbol{\Phi}_t\mathbf{S}_t)^{-1} + \mathbf{U}^\top\mathbf{C}_{xx}\xi_t(\mathbf{U}^\top\mathbf{C}_{xx}\boldsymbol{\Phi}_t\mathbf{S}_t)^{-1})}$$
$$\leq \frac{\|\boldsymbol{\Sigma}_\perp^2\mathbf{U}_\perp^\top\mathbf{C}_{xx}\boldsymbol{\Phi}_t(\mathbf{U}^\top\mathbf{C}_{xx}\boldsymbol{\Phi}_t)^{-1}\|_2 + (\|\mathbf{U}_\perp^\top\mathbf{C}_{xy}\eta_t\|_2 + \|\mathbf{U}_\perp^\top\mathbf{C}_{xx}\xi_t\|_2\|\mathbf{S}_t^{-1}\|_2)\|(\mathbf{U}^\top\mathbf{C}_{xx}\boldsymbol{\Phi}_t)^{-1}\|_2}{\sigma_{\min}(\boldsymbol{\Sigma}^2) - (\|\mathbf{U}^\top\mathbf{C}_{xy}\eta_t\|_2 + \|\mathbf{U}^\top\mathbf{C}_{xx}\xi_t\|_2\|\mathbf{S}_t^{-1}\|_2)\|(\mathbf{U}^\top\mathbf{C}_{xx}\boldsymbol{\Phi}_t)^{-1}\|_2}.$$
In the last inequality, we have that
$$\|(\mathbf{U}^\top\mathbf{C}_{xx}\boldsymbol{\Phi}_t)^{-1}\|_2 = \cos^{-1}\theta_{\max}(\boldsymbol{\Phi}_t,\mathbf{U}),$$
and
$$\begin{aligned}
\|\boldsymbol{\Sigma}_\perp^2\mathbf{U}_\perp^\top\mathbf{C}_{xx}\boldsymbol{\Phi}_t(\mathbf{U}^\top\mathbf{C}_{xx}\boldsymbol{\Phi}_t)^{-1}\|_2 &\leq \|\boldsymbol{\Sigma}_\perp^2\|_2\|\mathbf{U}_\perp^\top\mathbf{C}_{xx}\boldsymbol{\Phi}_t(\mathbf{U}^\top\mathbf{C}_{xx}\boldsymbol{\Phi}_t)^{-1}\|_2\\
&= \sigma_{k+1}^2\tan\theta_{\max}(\boldsymbol{\Phi}_t,\mathbf{U}),
\end{aligned}$$
$$\|\mathbf{U}^\top\mathbf{C}_{xx}\xi_t\|_2 \leq \|\xi_t\|_{\mathbf{C}_{xx}}, \qquad \|\mathbf{U}_\perp^\top\mathbf{C}_{xx}\xi_t\|_2 \leq \|\xi_t\|_{\mathbf{C}_{xx}},$$
$$\|\mathbf{U}^\top\mathbf{C}_{xy}\eta_t\|_2 \leq \sigma_1\|\eta_t\|_{\mathbf{C}_{yy}}, \qquad \|\mathbf{U}_\perp^\top\mathbf{C}_{xy}\eta_t\|_2 \leq \sigma_{k+1}\|\eta_t\|_{\mathbf{C}_{yy}},$$
where the last four inequalities are due to Eq. (1) as well as $\|\xi_t\|_{\mathbf{C}_{xx}} = \|\mathbf{C}_{xx}^{1/2}\xi_t\|_2$. In addition,
$$\begin{aligned}
\|\mathbf{S}_t^{-1}\|_2 &= \|(\tilde{\boldsymbol{\Psi}}_t^\top\mathbf{C}_{yy}\tilde{\boldsymbol{\Psi}}_t)^{1/2}\|_2 = \|\mathbf{C}_{yy}^{1/2}(\mathbf{C}_{yy}^{-1}\mathbf{C}_{xy}^\top\boldsymbol{\Phi}_t + \eta_t)\|_2\\
&\leq \sigma_1 + \|\eta_t\|_{\mathbf{C}_{yy}}.
\end{aligned}$$
We thus obtain that
$$\tan\theta_{\max}(\boldsymbol{\Phi}_{t+1},\mathbf{U})$$
$$\leq \frac{\sigma_{k+1}^2\tan\theta_{\max}(\boldsymbol{\Phi}_t,\mathbf{U}) + \dfrac{\sigma_{k+1}\|\eta_t\|_{\mathbf{C}_{yy}} + (\sigma_1 + \|\eta_t\|_{\mathbf{C}_{yy}})\|\xi_t\|_{\mathbf{C}_{xx}}}{\cos\theta_{\max}(\boldsymbol{\Phi}_t,\mathbf{U})}}{\sigma_k^2 - \dfrac{\sigma_1\|\eta_t\|_{\mathbf{C}_{yy}} + (\sigma_1 + \|\eta_t\|_{\mathbf{C}_{yy}})\|\xi_t\|_{\mathbf{C}_{xx}}}{\cos\theta_{\max}(\boldsymbol{\Phi}_t,\mathbf{U})}}$$
$$\leq \frac{\sigma_{k+1}^2\tan\theta_t + \dfrac{\sigma_{k+1}\|\eta_t\|_{\mathbf{C}_{yy}} + (\sigma_1 + \|\eta_t\|_{\mathbf{C}_{yy}})\|\xi_t\|_{\mathbf{C}_{xx}}}{\cos\theta_t}}{\sigma_k^2 - \dfrac{\sigma_1\|\eta_t\|_{\mathbf{C}_{yy}} + (\sigma_1 + \|\eta_t\|_{\mathbf{C}_{yy}})\|\xi_t\|_{\mathbf{C}_{xx}}}{\cos\theta_t}}$$
$$\leq \frac{\sigma_{k+1}^2 + \dfrac{\sigma_{k+1}\|\eta_t\|_{\mathbf{C}_{yy}} + (\sigma_1 + \|\eta_t\|_{\mathbf{C}_{yy}})\|\xi_t\|_{\mathbf{C}_{xx}}}{\sin\theta_t}}{\sigma_k^2 - \dfrac{\sigma_1\|\eta_t\|_{\mathbf{C}_{yy}} + (\sigma_1 + \|\eta_t\|_{\mathbf{C}_{yy}})\|\xi_t\|_{\mathbf{C}_{xx}}}{\cos\theta_t}} \cdot \tan\theta_t$$

$$\leq \quad \frac{\sigma_{k+1}^2 + (2\sigma_1 + \frac{\sigma_k^2 - \sigma_{k+1}^2}{12})\frac{\sigma_k^2 - \sigma_{k+1}^2}{12}}{\sigma_k^2 - (2\sigma_1 + \frac{\sigma_k^2 - \sigma_{k+1}^2}{12})\frac{\sigma_k^2 - \sigma_{k+1}^2}{12}} \tan\theta_t \qquad (\text{ by Eq. (2) })$$

$$\leq \quad \frac{\sigma_{k+1}^2 + \frac{\sigma_k^2 - \sigma_{k+1}^2}{4}}{\sigma_k^2 - \frac{\sigma_k^2 - \sigma_{k+1}^2}{4}} \tan\theta_t \qquad (\text{due to } \sigma_1 \leq 1)$$

$$= \quad \frac{\sigma_k^2 + 3\sigma_{k+1}^2}{3\sigma_k^2 + \sigma_{k+1}^2} \tan\theta_t$$

$$\leq \quad \exp\{-\frac{\sigma_k^2 - \sigma_{k+1}^2}{2\sigma_k^2}\} \tan\theta_t.$$

Similarly, we have

$$\tan\theta_{\max}(\boldsymbol{\Psi}_{t+1}, \mathbf{V}) \leq \exp\{-\frac{\sigma_k^2 - \sigma_{k+1}^2}{2\sigma_k^2}\} \tan\theta_t.$$

Taking the maximum over the left hand sides of the last two inequalities above, we arrive at

$$\tan\theta_{t+1} \leq \exp\{-\frac{\sigma_k^2 - \sigma_{k+1}^2}{2\sigma_k^2}\} \tan\theta_t,$$

and hence

$$\tan\theta_T \leq \exp\{-\frac{\sigma_k^2 - \sigma_{k+1}^2}{2\sigma_k^2} \cdot T\} \tan\theta_0 \triangleq \Xi,$$

Letting $\Xi = \epsilon$, i.e.,

$$T = O(\frac{\sigma_k^2}{\sigma_k^2 - \sigma_{k+1}^2} \log\frac{\tan\theta_0}{\epsilon}) = O(\frac{\sigma_k^2}{\sigma_k^2 - \sigma_{k+1}^2} \log\frac{1}{\epsilon\cos\theta_0}),$$

we obtain that $\tan\theta_T \leq \epsilon$. For subproblems, by Lemma 2, we have that

$$\log\frac{\epsilon_{t+1}(\tilde{\boldsymbol{\Phi}}_0)}{\epsilon_{t+1}(\tilde{\boldsymbol{\Phi}}_{t+1})} \quad = \quad \log\frac{2\epsilon_{t+1}(\tilde{\boldsymbol{\Phi}}_0)}{\|\xi_t\|_{\mathbf{C}_{xx},F}^2}$$

$$= \quad O(\log\frac{4k\sigma_1^2 \tan^2\theta_t}{(\frac{\sigma_k^2 - \sigma_{k+1}^2}{12} \min\{\sin\theta_t, \cos\theta_t\})^2})$$

$$= \quad O(\log\frac{\sigma_1}{\sigma_k^2 - \sigma_{k+1}^2} + \iota(\theta_t)),$$

where

$$\iota(\theta_t) = O(\log\max\{\frac{1}{\cos^2\theta_t}, \frac{\sin^2\theta_t}{\cos^4\theta_t}\}) = \begin{cases} O(\log\frac{1}{\cos\theta_0}), & \theta_t \text{ is large} \\ O(1), & \theta_t \text{ is small} \end{cases}.$$

The same equality holds for $\log\frac{\epsilon_{t+1}(\tilde{\boldsymbol{\Psi}}_0)}{\epsilon_{t+1}(\tilde{\boldsymbol{\Psi}}_{t+1})}$. Finally, following [1], a two-phase analysis of the running time based on $\theta_t$ yields the following total complexity

$$O(\frac{k\sigma_k^2}{\sigma_k^2 - \sigma_{k+1}^2} \mathrm{nnz}(\mathbf{X}, \mathbf{Y}) \kappa(\mathbf{X}, \mathbf{Y})(\log\frac{1}{\cos\theta_0} \log\frac{\sigma_1}{(\sigma_k^2 - \sigma_{k+1}^2)\cos\theta_0} +$$

$$\log\frac{1}{\epsilon} \log\frac{\sigma_1}{\sigma_k^2 - \sigma_{k+1}^2}) + \frac{dk^2\sigma_k^2}{\sigma_k^2 - \sigma_{k+1}^2} \log\frac{1}{\epsilon\cos\theta_0}),$$

where $\mathrm{nnz}(\mathbf{X}, \mathbf{Y}) = \mathrm{nnz}(\mathbf{X}) + \mathrm{nnz}(\mathbf{Y})$ and $\kappa(\mathbf{X}, \mathbf{Y}) = \max\{\kappa(\mathbf{C}_{xx}), \kappa(\mathbf{C}_{yy})\}$. $\qquad\square$

**Remark** Note that the recurrence equation (3) differs from those in [2] and [1] where $\boldsymbol{\Phi}_{t+1}$ is a function of $\boldsymbol{\Phi}_{t-1}$ instead of $\boldsymbol{\Phi}_t$.

**Theorem 2** Given data matrices $\mathbf{X}$ and $\mathbf{Y}$, FastTALS-CCA computes a $d_x \times k$ matrix $\mathbf{\Phi}_T$ and a $d_y \times k$ matrix $\mathbf{\Psi}_T$ which are estimates of top-$k$ canonical subspaces $(\mathbf{U}, \mathbf{V})$ with an error of $\epsilon$, i.e., $\mathbf{\Phi}_T^\top \mathbf{C}_{xx} \mathbf{\Phi}_T = \mathbf{\Psi}_T^\top \mathbf{C}_{yy} \mathbf{\Psi}_T = \mathbf{I}$ and $\tan \theta_T \le \epsilon$, in $T = O(\frac{\sigma_k^2 - c\sigma_1 \beta}{\sigma_k^2 - \sigma_{k+1}^2 - 4c\sigma_1 \beta} \log \frac{1}{\epsilon \cos \theta_0})$ iterations if $\theta_0 \le \frac{\pi}{4}$. If Nesterov's accelerated gradient descent is used as the least-squares solver, the running time is at most

$$O(\frac{k(\sigma_k^2 - c\sigma_1 \beta)}{\sigma_k^2 - \sigma_{k+1}^2 - 4c\sigma_1 \beta} \mathrm{nnz}(\mathbf{X}, \mathbf{Y})\kappa(\mathbf{X}, \mathbf{Y})(\log \frac{1}{\cos \theta_0} \log \frac{\sigma_1}{(\sigma_k^2 - \sigma_{k+1}^2)\cos \theta_0} +$$

$$\log \frac{1}{\epsilon} \log \frac{\sigma_1}{\sigma_k^2 - \sigma_{k+1}^2}) + \frac{k^2(\sigma_k^2 - c\sigma_1 \beta)}{\sigma_k^2 - \sigma_{k+1}^2 - 4c\sigma_1 \beta} \max\{d_x, d_y\} \log \frac{1}{\epsilon \cos \theta_0}),$$

where $c > 0$ is a constant.

**Proof** We only give key steps of the proof as the remainder is the same as above. Let $\tilde{\theta}_t \triangleq \theta_{\max}(\mathbf{\Phi}_t, \mathbf{U})$. The coupled and inexact update equations are as follows:

$$\begin{cases} \tilde{\mathbf{\Phi}}_{t+1} = \mathbf{C}_{xx}^{-1}\mathbf{C}_{xy}\mathbf{\Psi}_t - \beta\mathbf{\Phi}_{t-1} + \xi_t, & \mathbf{\Phi}_{t+1} = \tilde{\mathbf{\Phi}}_{t+1}\mathbf{R}_{t+1} \\ \tilde{\mathbf{\Psi}}_{t+1} = \mathbf{C}_{yy}^{-1}\mathbf{C}_{xy}^\top\mathbf{\Phi}_{t+1} - \beta\mathbf{\Psi}_t + \eta_{t+1}, & \mathbf{\Psi}_{t+1} = \tilde{\mathbf{\Psi}}_{t+1}\mathbf{S}_{t+1} \end{cases}.$$

We then have that

$$\mathbf{\Phi}_{t+1} = (\mathbf{C}_{xx}^{-1}\mathbf{C}_{xy}(\mathbf{C}_{yy}^{-1}\mathbf{C}_{xy}^\top\mathbf{\Phi}_t - \beta\mathbf{\Psi}_{t-1} + \eta_t)\mathbf{S}_t - \beta\mathbf{\Phi}_{t-1} + \xi_t)\mathbf{R}_{t+1}. \tag{4}$$

One then gets that

$$\mathbf{U}_\perp^\top \mathbf{C}_{xx}\mathbf{\Phi}_{t+1} = (\mathbf{\Sigma}_\perp^2 \mathbf{U}_\perp^\top\mathbf{C}_{xx}\mathbf{\Phi}_t\mathbf{S}_t - \beta\mathbf{U}_\perp^\top\mathbf{C}_{xy}\mathbf{\Psi}_{t-1} + \mathbf{U}_\perp^\top\mathbf{C}_{xy}\eta_t\mathbf{S}_t - \beta\mathbf{U}_\perp^\top\mathbf{C}_{xx}\mathbf{\Phi}_{t-1} + \mathbf{U}_\perp^\top\mathbf{C}_{xx}\xi_t)\mathbf{R}_{t+1},$$

$$\mathbf{U}^\top \mathbf{C}_{xx}\mathbf{\Phi}_{t+1} = (\mathbf{\Sigma}^2 \mathbf{U}^\top\mathbf{C}_{xx}\mathbf{\Phi}_t\mathbf{S}_t - \beta\mathbf{U}^\top\mathbf{C}_{xy}\mathbf{\Psi}_{t-1} + \mathbf{U}^\top\mathbf{C}_{xy}\eta_t\mathbf{S}_t - \beta\mathbf{U}^\top\mathbf{C}_{xx}\mathbf{\Phi}_{t-1} + \mathbf{U}^\top\mathbf{C}_{xx}\xi_t)\mathbf{R}_{t+1}.$$

There is a certain numerical constant $c$ such that $\sin \tilde{\theta}_{t-1} \le c \sin \tilde{\theta}_t$ for a finite $t$. We can write that

$$\tan \tilde{\theta}_{t+1}$$

$$\le \frac{\|(\mathbf{\Sigma}_\perp^2 \mathbf{U}_\perp^\top\mathbf{C}_{xx}\mathbf{\Phi}_t\mathbf{S}_t - \beta\mathbf{U}_\perp^\top\mathbf{C}_{xy}\mathbf{\Psi}_{t-1} + \mathbf{U}_\perp^\top\mathbf{C}_{xy}\eta_t\mathbf{S}_t - \beta\mathbf{U}_\perp^\top\mathbf{C}_{xx}\mathbf{\Phi}_{t-1} + \mathbf{U}_\perp^\top\mathbf{C}_{xx}\xi_t)(\mathbf{U}^\top\mathbf{C}_{xx}\mathbf{\Phi}_t\mathbf{S}_t)^{-1}\|_2}{\sigma_{\min}(\mathbf{\Sigma}^2 + (-\beta\mathbf{U}^\top\mathbf{C}_{xy}\mathbf{\Psi}_{t-1} + \mathbf{U}^\top\mathbf{C}_{xy}\eta_t\mathbf{S}_t - \beta\mathbf{U}^\top\mathbf{C}_{xx}\mathbf{\Phi}_{t-1} + \mathbf{U}^\top\mathbf{C}_{xx}\xi_t)(\mathbf{U}^\top\mathbf{C}_{xx}\mathbf{\Phi}_t\mathbf{S}_t)^{-1})}$$

$$\le \frac{\sigma_{k+1}^2 \tan \tilde{\theta}_t + \dfrac{\beta\sigma_{k+1} \sin \tilde{\theta}_{t-1} + \beta(\sigma_1 + \beta + \|\eta_t\|_{\mathbf{C}_{yy}}) \sin \tilde{\theta}_{t-1} + \sigma_{k+1}\|\eta_t\|_{\mathbf{C}_{yy}} + (\sigma_1 + \beta + \|\eta_t\|_{\mathbf{C}_{yy}})\|\xi_t\|_{\mathbf{C}_{xx}}}{\cos \tilde{\theta}_t}}{\sigma_k^2 - \dfrac{\beta\sigma_1 \sin \tilde{\theta}_{t-1} + \beta(\sigma_1 + \beta + \|\eta_t\|_{\mathbf{C}_{yy}}) \sin \tilde{\theta}_{t-1} + \sigma_1\|\eta_t\|_{\mathbf{C}_{yy}} + (\sigma_1 + \beta + \|\eta_t\|_{\mathbf{C}_{yy}})\|\xi_t\|_{\mathbf{C}_{xx}}}{\cos \tilde{\theta}_t}}$$

$$\le \frac{\sigma_{k+1}^2 \tan \tilde{\theta}_t + \dfrac{c\beta\sigma_{k+1} \sin \tilde{\theta}_t + c\beta(\sigma_1 + \beta + \|\eta_t\|_{\mathbf{C}_{yy}}) \sin \tilde{\theta}_t + \sigma_{k+1}\|\eta_t\|_{\mathbf{C}_{yy}} + (\sigma_1 + \beta + \|\eta_t\|_{\mathbf{C}_{yy}})\|\xi_t\|_{\mathbf{C}_{xx}}}{\cos \tilde{\theta}_t}}{\sigma_k^2 - \dfrac{c\beta\sigma_1 \sin \tilde{\theta}_t + c\beta(\sigma_1 + \beta + \|\eta_t\|_{\mathbf{C}_{yy}}) \sin \tilde{\theta}_t + \sigma_1\|\eta_t\|_{\mathbf{C}_{yy}} + (\sigma_1 + \beta + \|\eta_t\|_{\mathbf{C}_{yy}})\|\xi_t\|_{\mathbf{C}_{xx}}}{\cos \tilde{\theta}_t}}$$

$$\le \frac{\sigma_{k+1}^2 + c\beta\sigma_{k+1} + c\beta(\sigma_1 + \beta + \dfrac{\sigma_k^2 - \sigma_{k+1}^2}{12}) + (2\sigma_1 + \dfrac{\sigma_k^2 - \sigma_{k+1}^2}{12})\dfrac{\sigma_k^2 - \sigma_{k+1}^2}{12}}{\sigma_k^2 - c\beta\sigma_{k+1} \tan \tilde{\theta}_{t-1} - c\beta(\sigma_1 + \beta + \dfrac{\sigma_k^2 - \sigma_{k+1}^2}{12}) \tan \tilde{\theta}_{t-1} - (2\sigma_1 + \dfrac{\sigma_k^2 - \sigma_{k+1}^2}{12})\dfrac{\sigma_k^2 - \sigma_{k+1}^2}{12}} \tan \theta_t$$

$$\le \frac{\sigma_{k+1}^2 + 4c\beta\sigma_1 + \dfrac{\sigma_k^2 - \sigma_{k+1}^2}{4}}{\sigma_k^2 - 4c\beta\sigma_1 - \dfrac{\sigma_k^2 - \sigma_{k+1}^2}{4}} \tan \theta_t$$

$$= \frac{\sigma_k^2 + 3\sigma_{k+1}^2 + 16c\beta\sigma_1}{3\sigma_k^2 + \sigma_{k+1}^2 - 16c\beta\sigma_1} \tan \theta_t$$

$$\le \exp\{-\frac{\sigma_k^2 - \sigma_{k+1}^2 - 16c\beta\sigma_1}{2\sigma_k^2 - 8c\beta\sigma_1}\} \tan \theta_t.$$

$\square$

**Lemma 1** [1]

$$\sin\theta_{\max}(\mathbf{\Phi}, \mathbf{U}) = \|\mathbf{U}_\perp^\top \mathbf{C}_{xx}\mathbf{\Phi}\|_2 \quad \text{and} \quad \tan\theta_{\max}(\mathbf{\Phi}, \mathbf{U}) = \|\mathbf{U}_\perp^\top \mathbf{C}_{xx}\mathbf{\Phi}(\mathbf{U}^\top \mathbf{C}_{xx}\mathbf{\Phi})^{-1}\|_2$$

if $\mathbf{U}^\top \mathbf{C}_{xx}\mathbf{\Phi}$ is invertible.

**Lemma 2** For the least-squares subproblem, let $\mathbf{\Phi}_t^* \triangleq \arg\min l_t(\mathbf{\Phi}) = \mathbf{C}_{xx}^{-1}\mathbf{C}_{xy}\mathbf{\Psi}_{t-1}$ and $\epsilon_t(\mathbf{\Phi}) = l_t(\mathbf{\Phi}) - l_t(\mathbf{\Phi}_t^*)$. We then have $\epsilon_t(\mathbf{\Phi}) = \frac{1}{2}\|\mathbf{\Phi} - \mathbf{\Phi}_t^*\|_{\mathbf{C}_{xx},F}^2$. Particularly, $\epsilon_t(\tilde{\mathbf{\Phi}}_0) \leq 2k\sigma_1^2\tan^2\theta_{t-1}$, where $\|\mathbf{A}\|_{\mathbf{\Lambda},F} = \|\mathbf{\Lambda}^{1/2}\mathbf{A}\|_F$ and $\tilde{\mathbf{\Phi}}_0 = \mathbf{\Phi}_{t-1}(\mathbf{\Phi}_{t-1}^\top \mathbf{C}_{xx}\mathbf{\Phi}_{t-1})^{-1}(\mathbf{\Phi}_{t-1}^\top \mathbf{C}_{xy}\mathbf{\Psi}_{t-1})$. In addition, Nesterov's accelerated gradient descent takes $O(\mathrm{nnz}(\mathbf{Y}) + \mathrm{nnz}(\mathbf{X})\sqrt{\kappa(\mathbf{C}_{xx})}\log\frac{\epsilon_t(\tilde{\mathbf{\Phi}}_0)}{\epsilon_t(\tilde{\mathbf{\Phi}}_t)})$ complexity to reach sub-optimality $\epsilon_t(\tilde{\mathbf{\Phi}}_t)$.

**Proof** Noting that $l_t(\mathbf{\Phi}_t^*) = -\frac{1}{2}\mathrm{tr}(\mathbf{\Psi}_{t-1}^\top \mathbf{C}_{xy}^\top \mathbf{C}_{xx}^{-1}\mathbf{C}_{xy}\mathbf{\Psi}_{t-1}) + \frac{1}{2n}\|\mathbf{Y}^\top \mathbf{\Psi}_{t-1}\|_F^2$, we have that

$$\begin{aligned}
\frac{1}{2}\|\mathbf{\Phi} - \mathbf{\Phi}_t^*\|_{\mathbf{C}_{xx},F}^2 &= \frac{1}{2}\mathrm{tr}((\mathbf{\Phi} - \mathbf{\Phi}_t^*)^\top \mathbf{C}_{xx}(\mathbf{\Phi} - \mathbf{\Phi}_t^*)) \\
&= \mathrm{tr}(\frac{1}{2}\mathbf{\Phi}^\top \mathbf{C}_{xx}\mathbf{\Phi} - \mathbf{\Phi}^\top \mathbf{C}_{xx}\mathbf{\Phi}_t^* + \frac{1}{2}(\mathbf{\Phi}_t^*)^\top \mathbf{C}_{xx}\mathbf{\Phi}_t^*) \\
&= \mathrm{tr}(\frac{1}{2}\mathbf{\Phi}^\top \mathbf{C}_{xx}\mathbf{\Phi} - \mathbf{\Phi}^\top \mathbf{C}_{xx}\mathbf{\Psi}_{t-1} + \frac{1}{2}\mathbf{\Psi}_{t-1}^\top \mathbf{C}_{xy}^\top \mathbf{C}_{xx}\mathbf{C}_{xy}\mathbf{\Psi}_{t-1}) \\
&= l_t(\mathbf{\Phi}) - l_t(\mathbf{\Phi}_t^*) = \epsilon_t(\mathbf{\Phi}).
\end{aligned}$$

Let $h_t(\mathbf{\Gamma}) = l_t(\mathbf{\Phi}\mathbf{\Gamma}) - l_t(\mathbf{\Phi}_t^*)$. Setting $\frac{\partial}{\partial\mathbf{\Gamma}}h_t(\mathbf{\Gamma}) = \mathbf{\Phi}_{t-1}^\top \mathbf{C}_{xx}\mathbf{\Phi}_{t-1}\mathbf{\Gamma} - \mathbf{\Phi}_{t-1}^\top \mathbf{C}_{xy}\mathbf{\Psi}_{t-1} = 0$ yields the optimal $\mathbf{\Gamma}^\star = (\mathbf{\Phi}_{t-1}^\top \mathbf{C}_{xx}\mathbf{\Phi}_{t-1})^{-1}\mathbf{\Phi}_{t-1}^\top \mathbf{C}_{xy}\mathbf{\Psi}_{t-1}$. That is, $\tilde{\mathbf{\Phi}}_0 = \mathbf{\Phi}_{t-1}\mathbf{\Gamma}^\star$. If we use $\tilde{\mathbf{\Gamma}}$ such that $\mathbf{U}^\top \mathbf{C}_{xx}\mathbf{\Phi}_{t-1}\tilde{\mathbf{\Gamma}} - \mathbf{U}^\top \mathbf{C}_{xy}\mathbf{\Psi}_{t-1} = 0$, i.e.,

$$\tilde{\mathbf{\Gamma}} = (\mathbf{U}^\top \mathbf{C}_{xx}\mathbf{\Phi}_{t-1})^{-1}\mathbf{U}^\top \mathbf{C}_{xy}\mathbf{\Psi}_{t-1} = (\mathbf{U}^\top \mathbf{C}_{xx}\mathbf{\Phi}_{t-1})^{-1}\mathbf{\Sigma}\mathbf{V}^\top \mathbf{C}_{yy}\mathbf{\Psi}_{t-1},$$

we then have that

$$\begin{aligned}
\epsilon_t(\tilde{\mathbf{\Phi}}_0) &\leq \epsilon_t(\mathbf{\Phi}_{t-1}\tilde{\mathbf{\Gamma}}) \\
&= \frac{1}{2}\|\mathbf{\Phi}_{t-1}\tilde{\mathbf{\Gamma}} - \mathbf{\Phi}_t^*\|_{\mathbf{C}_{xx},F}^2 \\
&= \frac{1}{2}(\|\mathbf{U}^\top \mathbf{C}_{xx}(\mathbf{\Phi}_{t-1}\tilde{\mathbf{\Gamma}} - \mathbf{C}_{xx}^{-1}\mathbf{C}_{xy}\mathbf{\Psi}_{t-1})\|_F^2 + \|\mathbf{U}_\perp^\top \mathbf{C}_{xx}(\mathbf{\Phi}_{t-1}\tilde{\mathbf{\Gamma}} - \mathbf{C}_{xx}^{-1}\mathbf{C}_{xy}\mathbf{\Psi}_{t-1})\|_F^2) \\
&= \frac{1}{2}\|\mathbf{U}_\perp^\top \mathbf{C}_{xx}(\mathbf{\Phi}_{t-1}\tilde{\mathbf{\Gamma}} - \mathbf{C}_{xx}^{-1}\mathbf{C}_{xy}\mathbf{\Psi}_{t-1})\|_F^2 = \frac{1}{2}\|\mathbf{U}_\perp^\top \mathbf{C}_{xx}\mathbf{\Phi}_{t-1}\tilde{\mathbf{\Gamma}} - \mathbf{U}_\perp^\top \mathbf{C}_{xy}\mathbf{\Psi}_{t-1}\|_F^2 \\
&= \frac{1}{2}\|\mathbf{U}_\perp^\top \mathbf{C}_{xx}\mathbf{\Phi}_{t-1}\tilde{\mathbf{\Gamma}} - \mathbf{\Sigma}_\perp^\top \mathbf{V}_\perp^\top \mathbf{C}_{yy}\mathbf{\Psi}_{t-1}\|_F^2 \qquad (\text{ by Equation(1) }) \\
&\leq k(\|\mathbf{U}_\perp^\top \mathbf{C}_{xx}\mathbf{\Phi}_{t-1}\|_2^2\|\tilde{\mathbf{\Gamma}}\|_2^2 + \|\mathbf{\Sigma}_\perp\|_2^2\|\mathbf{V}_\perp^\top \mathbf{C}_{yy}\mathbf{\Psi}_{t-1}\|_2^2) \\
&\leq k(\frac{\sigma_1^2\sin^2\theta_{\max}(\mathbf{\Phi}_{t-1}, \mathbf{U})}{\cos^2\theta_{\max}(\mathbf{\Phi}_{t-1}, \mathbf{U})} + \sigma_{k+1}^2\sin^2\theta_{\max}(\mathbf{\Psi}_{t-1}, \mathbf{V})) \\
&\leq k(\sigma_1^2\tan^2\theta_{\max}(\mathbf{\Phi}_{t-1}, \mathbf{U}) + \sigma_{k+1}^2\tan^2\theta_{\max}(\mathbf{\Psi}_{t-1}, \mathbf{V})) \\
&\leq 2k\sigma_1^2\tan^2\theta_{t-1}.
\end{aligned}$$

The proof completes by noting that $l_t(\mathbf{\Phi})$ is $\lambda_{\max}(\mathbf{C}_{xx})$-smooth and $\lambda_{\min}(\mathbf{C}_{xx})$-strongly convex. $\square$

**Additional Experiments**   The convergence curves of all the ALS algorithms in terms of $(f^\star - f)/f^\star \triangleq (\mathrm{tr}(\boldsymbol{\Sigma}) - \mathrm{tr}(\boldsymbol{\Phi}_t^\top \mathbf{C}_{xy} \boldsymbol{\Psi}_t))/\mathrm{tr}(\boldsymbol{\Sigma})$ are given in Figure 1. Figure 2 shows the comparison of the ALS algorithms with the shift-and-invert preconditioning based method in the vector setting. Figure 3 reports the performance of the ALS algorithms with varying block sizes. Figure 4 shows the convergence results on the Youtube dataset.

Figure 1: Performance of the ALS algorithms in terms of $(f^\star - f)/f^\star$.

Figure 2: Comparison with the shift-and-invert preconditioning based methods.

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

Figure 3: Performance of the ALS algorithms with varying block sizes.

Figure 4: Performance of the ALS algorithms on Youtube.