[Reviews · NeurIPS 2019]

Reviewer 1



It is interesting to see that CCA is attracting attention again. Given the noticeable speed up achievable with the methods in this paper, it should be of interest to people working on (applications of) CCA. What I found missing was a better or more in-depth explanation/motivation as to where equation (4) comes from. How was it derived? Also, but this is a minor point, the lack of convergence analysis is indeed weakening this contribution. In section 6, the authors convincingly justify this but any attempt in this direction would considerably strengthen this paper.

Reviewer 2



The paper propose the truly and in exactly alternating least squares. Instead of approximately solving two independent linear systems, in each iteration, the algorithm solves two coupled linear systems of half the size. The submission is clearly written, and well structured. Setting up the premise, reviewing related work and motivation for the authors proposed algorithm. The mathematical derivations appear correct and algorithm runtime complexity is provided (proofs in supplementary material). The proposed algorithm and results are in my view significant, showing the ability to cut run time while retaining quality measures. The experiments section could be expanded to further detail the results. E.g. providing some views on the difference of performance (where existing methods fail to work or unable to find a solution). Likewise, in the experiments (or discussion) section, a view as to whether the proposed algorithm would work in all cases or a set of them (indication for possible future work).

Reviewer 3



SUMMARY: The paper presents a technical improvement to the alternating least-squares (ALS) solution to CCA. The novel method is named “Truly and Inexact Alternating Least Squares” (TALS). The idea is to solve two coupled linear systems of half size per iteration. In previous work, two independent linear systems have been solved which deteriorates the final canonical correlation between the views due to inaccurate assignment of the weights on the variables. Additionally, a faster version, FastTALS, is proposed. FastTALS applies momentum acceleration to increase the speed of the optimisation. The momentum hyperparameter requires tuning, and in case hand-tuning is not preferred, an adaptive version of FastTALS is proposed. It automatically sets the tuning hyperparameter, during optimisation. ORIGINALITY: It is a novel idea to couple together two linear systems of half size compared to the two previous approaches. In the paper, it is clearly demonstrated what the two previous approaches and the proposed method look like. The literature review is adequate and well-referenced. QUALITY: The theory presented is complete and proved. The presented method and its advanced versions are empirically assessed on three real-world datasets: Mediamill, MNIST, and JW11. The proposed methods are compared with the two earlier approaches. The performances of the methods are assessed using a (novel?) measure by computing the squared sine value of the largest principal angle between the learnt weight matrix and the ground truth obtained using MATLAB’s svds function. In other words, the performance is evaluated in terms of convergence to the ground truth U and V. The convergence to U and V is shown as a function of the time and numbers of passes through the data. More experiments are reported in the supplementary material. Overall, the proposed approach is evaluated both theoretically and empirically. CLARITY: The paper is clearly written and well-organised. The MATLAB implementations of TALS and FastTALS are accessible through an anonymous dropbox link, so the presented results can possibly be reproduced. SIGNIFICANCE: The proposed approach seems to give both a significant speed-up to ALS-based CCA and improve the accuracy of the results. For practitioners, it is important that the canonical weights are correct so that the relations between the variables can be inferred. Therefore, this approach would be recommended instead of the previous ones. *** AFTER REBUTTAL*** I have read the authors' response and other reviewers' comments. My score remains the same. ***END OF COMMENT***

[Author Response · NeurIPS 2019]

We appreciate reviewers' comments. Below are our responses to each reviewer.

**Reviewer 1**  1) About the motivation of Equation (4). Consider the case of exact updates of two canonical variables
$\mathbf{\Phi}$ and $\mathbf{\Psi}$ in Equation (3), i.e., $\xi_t = \eta_t = \mathbf{0}$. In the second line of Equation (3), it is easy to see that if $\mathbf{\Phi}_t$ is equal to
the canonical subspace $\mathbf{\Phi}^\star = \mathbf{C}_{xx}^{-1/2}\mathbf{U}$ (i.e., ground truth) then $\tilde{\mathbf{\Phi}}_{t+1} = \mathbf{C}_{yy}^{-1/2}\mathbf{V}\mathbf{\Sigma}$ will span the canonical subspace
$\mathbf{\Psi}^\star = \mathbf{C}_{yy}^{-1/2}\mathbf{V}$. This basically means that if $\mathbf{\Phi}_t$ is closer to the ground truth, $\mathbf{\Psi}_{t+1}$ will be closer to its ground truth as
well. This in turn suggests that replacing $\mathbf{\Phi}_t$ with $\mathbf{\Phi}_{t+1}$ in the second line of Equation (3) may improve the convergence,
because $\mathbf{\Phi}_{t+1}$ is supposed to be closer to the ground truth than $\mathbf{\Phi}_t$. This replacement makes Equation (3) have deviated
from the standard power iteration from which Equation (3) is derived. Thus it is no longer necessary for us to stick
to the joint orthogonalization of two canonical variables. Instead, sperate orthogonalizations are used. The proof of
Theorem 3.1 justifies this change. We then can arrive at Equation (4) in the inexact case.

2) About the lack of convergence analysis. We guess the reviewer referred to the tight convergence analysis of FastTALS,
as TALS is globally convergent and the analysis is tight (i.e., rate matching the method) as stated in Theorem 3.1.
FastTALS is locally convergent and the analysis is not tight as stated in Theorem 4.1. First, the global convergence
actually is not an issue, because one can use our globally convergent TALS algorithm to warm start FastTALS so that
two canonical variables are sufficiently close to the ground truth. On the tight analysis, this is indeed difficult. We
tried to follow the work of accelerated stochastic power method by Xu et al., 2018. for a tight analysis. However, their
analysis only considered the vector case $k = 1$ in the stochastic setting where there are special structures, e.g., the
quantity inside the trace in Problem (1) is a scalar, that can be sufficiently utilized. In our case, this quantity is a matrix
and many analysis tricks fail to be applied. The extensions from vector to block are often difficult for this class of
problems. More difficulties arise in our case due to the coupling of update equations as well as approximation errors.
We thus leave it to our future work at the current stage.

3) We will follow the suggestion to improve the readability and move the description of the datasets back.

**Reviewer 2**  1) View on the difference of performance. First, algorithms ALS-$k$ (using block size $k$ and adapted
from ALS for vector setting $k = 1$) and CCALin-$k$ (using block size $k$ and adapted from CCALin for block setting)
are introduced to show the necessity of using block size $2k$ in order for them to recover top-$k$ canonical subspaces.
Throughout our experiments, they indeed fail to learn anything because their ground truth do not cover the top-
$k$ canonical subspaces.  Second, the reason why CCALin is worse than our algorithms (TALS, FastTALS, and
AdaFastTALS) is that CCALin needs to use block size $2k$ and a post-processing step that randomly projects the resulting
$2k$-dimensional subspaces onto $k$-dimensional subspaces.

2) View as to whether the proposed algorithm would work in all cases or a set of them. One premise of our algorithms
is that they work in the offline setting, i.e., the data pair $(\mathbf{X}, \mathbf{Y})$ is ready. This means that our algorithms may not work
in streaming/online setting directly. However, following the idea of GenOja (Kush Bhatia et al. NeurIPS 2018), we may
use one step of stochastic gradient descent as the least-squares solver. This might give rise to new algorithms, i.e., truly
streaming versions of our algorithms, and is well worth investigating. We may also consider our algorithms in robust
settings for future work.

3) Applications to downstream tasks. This is a good suggestion for the extension of the present work.

**Reviewer 3**  1) Simulation study. We initially planned simulation study. However, we soon found that there is no
simple way to generate the simulated data $(\mathbf{X}, \mathbf{Y})$ using $\mathbf{U}$ and $\mathbf{V}$. This is because we can only use the singular value
decomposition $(\frac{1}{n}\mathbf{X}\mathbf{X}^\top)^{-1/2}\frac{1}{n}\mathbf{X}\mathbf{Y}^\top(\frac{1}{n}\mathbf{Y}\mathbf{Y}^\top)^{-1/2} = \mathbf{C}_{xx}^{-1/2}\mathbf{C}_{xy}\mathbf{C}_{yy}^{-1/2} = \mathbf{C} = \mathbf{U}\mathbf{\Sigma}\mathbf{V}^\top$ to generate $\mathbf{C}$ from random
matrices $\mathbf{U}$, $\mathbf{\Sigma}$, and $\mathbf{V}$, but cannot recover $\mathbf{X}$ and $\mathbf{Y}$ from $\mathbf{C}$. This may be the reason why there are no previous CCA
works that did experiments on such simulated data (to the best of our knowledge). Nonetheless, we did experiments on a
randomly generated data pair $(\mathbf{X}, \mathbf{Y})$ (the first two plots in the figure) for which the ground truth is obtained by matlab's
function svds. Contrastingly, the performance improvements of our algorithms on real data are more prominent.

2) Convergence of competing methods. As mentioned in the first item of our response to Reviewer 2, ALS-$k$ and
CCALin-$k$ fail to recover top-$k$ canonical subspaces for CCA. Given sufficient running time, CCALin indeed can
recover top-$k$ canonical subspaces, as shown in the last two plots of the figure.



[Meta-Review · NeurIPS 2019]

The paper puts forward improved alternating least squares optimization for canonical correlation analysis by introducing a coupling between the subproblems that speeds up convergence, as well as an faster accelerated optimization. The reviewers found the contributions significant both in theory and practice and the paper sound and well-written.